# Scaling Large Motion Models with Million-Level Human Motions

**Ye Wang** [* 1]  **Sipeng Zheng** [* 2]  **Bin Cao** [2 3]  **Qianshan Wei** [4]  **Weishuai Zeng** [5]  **Qin Jin** [1]  **Zongqing Lu** [† 5 6]

## Abstract

Inspired by the recent success of LLMs, the field of human motion understanding has increasingly shifted toward developing large motion models. Despite some progress, current efforts remain far from achieving truly generalist models, primarily due to the lack of massive high-quality data. To address this gap, we present MotionLib, the first million-level dataset for motion generation, which is at least $15\times$ larger than existing counterparts and enriched with hierarchical text descriptions. Using MotionLib, we train a large motion model named Being-M0, demonstrating robust performance across a wide range of human activities, including unseen ones. Through systematic investigation, for the first time, we highlight the importance of scaling both data and model size for advancing motion generation, along with key insights to achieve this goal. To better integrate the motion modality, we propose Motionbook, an innovative motion encoding approach including (1) a compact yet lossless feature to represent motions; (2) a novel 2D lookup-free motion tokenizer that preserves fine-grained motion details while expanding codebook capacity, significantly enhancing the representational power of motion tokens. We believe this work lays the groundwork for developing more versatile and powerful motion generation models in the future. For further details, visit https://beingbeyond.github.io/Being-M0/.

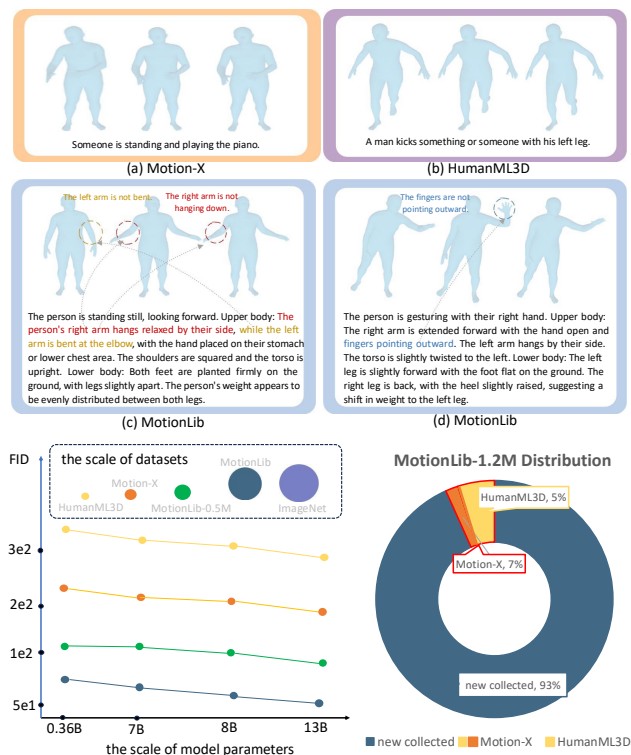

Figure 1: **TOP**: While existing models perform well on small-scale datasets like Motion-X and HumanML3D, they struggle with out-of-domain concepts on MotionLib, exhibiting limited generalization. **DOWN**: Curves showing the effects of scaling up large motion models. MotionLib is the first large T2M dataset comparable in scale to visual benchmarks like ImageNet.

## 1. Introduction

Motion generation is an emerging field with a wide range of applications in video games, filmmaking, and robotics. At the forefront of this area is text-to-motion generation (T2M) (Ahn et al., 2018; Ahuja & Morency, 2019), which plays a crucial role in translating natural language into human motions. State-of-the-art (SoTA) T2M models typically rely on a combination of the motion tokenizer (e.g., vector quantization VQ (Van Den Oord et al., 2017)), along with a text encoder (e.g., CLIP (Radford et al., 2021)) and decoder (e.g., GPT-2 (Radford et al., 2019)) to generate motion sequences from text commands. Despite the availability of

---

[*]Equal contribution  [1]Renmin University of China [2]Beijing Academy of Artificial Intelligence [3]Institute of Automation, Chinese Academy of Sciences [4]Southeast University [5]Peking University [6]BeingBeyond. Correspondence to: Zongqing Lu <zongqing.lu@pku.edu.cn>.

*Proceedings of the $42^{nd}$ International Conference on Machine Learning*, Vancouver, Canada. PMLR 267, 2025. Copyright 2025 by the author(s).

a few high-quality datasets (Guo et al., 2022a; Lin et al., 2024) curated in recent years, their limited size restricts current methods to a narrow range of scenarios, creating performance bottlenecks when addressing diverse or unseen motions, as illustrated in Figure 1 (TOP).

Recently, the rapid advancement of large language models (LLMs) in multimodal learning has been significantly bolstered by the availability of vast data. In contrast, the volume of motion data remains considerably smaller than that of visual-text data, as shown in Figure 1 (DOWN). This disparity primarily arises from the high costs associated with motion data collection, which often requires specialized wearable devices and substantial human labor for annotation. Consequently, developing a SoTA large motion model based on LLMs presents a significant challenge and remains an unresolved issue. While some recent efforts (Jiang et al., 2023) have explored this direction, the effectiveness of large motion models has yet to be fully demonstrated.

In this paper, we aim to address the question: "*Can scaling the large motion model and data benefit motion generation?*" To tackle this, we develop a systematic data collection pipeline to build MotionLib, the first large-scale dataset containing over 1.2M motion sequences — at least 15× larger than current counterparts. This initiative provides a solid foundation for building robust, universally applicable motion models and offers a comprehensive testbed for future research. Using MotionLib, we conduct a comprehensive investigation into the large motion model. For the first time, we show the scaling law of both data and model size in motion generation, which significantly reduces joint prediction errors while improving generalization to novel motions. In addition, this research identifies several key factors driving their advancement and offers valuable insights for future model design (e.g., LoRA or full-parameter tuning).

In addition, we argue that large motion models are constrained by inadequate motion representations. **First**, most approaches transform motion into discrete tokens via VQ, which are then processed by autoregressive models to generate motion sequences. While these methods have achieved impressive results, they suffer from two major drawbacks: **(1) Information Loss**: The VQ process inevitably discards critical motion details. Given a motion sequence with $D$-dimensional features $\mathcal{M} = \{m_1, m_2, ..., m_T\}$, where $m_i \in \mathbb{R}^D$, VQ compresses it into a sequence of 1D embeddings of size $\lfloor T/\alpha \rfloor \times d$, where $\alpha$ is the temporal downsampling ratio and $d$ is the codebook dimension. Unlike images, which consist of uniform RGB pixel values, each motion state $m_i$ encodes diverse features (e.g., joint position, velocity, foot-ground contact). Representing such complex states with a single 1D embedding is insufficient, leading to information loss and restricting the model's ability to generate motion at a fine-grained, part-level resolution.

**(2) Limited Codebook Size:** Existing VQs rely on small codebooks, meaning that all generated motions must be selected from a constrained set of predefined options. As a result, these 1D embeddings fail to capture the full diversity of human motion. **Second**, most works adopt the H3D-format feature (Guo et al., 2022a) to represent motion, which omits crucial information (e.g., original rotation) and requires time-consuming methods (Bogo et al., 2016) to recover. This inefficiency hinders real-world applications (e.g., animation), where real-time generation is essential.

To address these issues, we propose MotionBook, a novel motion encoding approach. Specifically, MotionBook first chooses a more compact yet lossless feature to represent $\mathcal{M}$, which preserves motion information without compromising performance. Additionally, it introduces a new motion quantization method, 2D-LFQ. Our 2D-LFQ reformulates each motion sequence as a 2D image with a single channel, represented as $\mathcal{M} \in R^{T \times D \times 1}$. By expanding the motion sequence's dimensionality from 1D to 2D, we enhance the encoder's capacity, enabling it to capture complex motion patterns while retaining more critical details after tokenization. While increasing the codebook size is a straightforward way to enhance expressiveness, it often leads to codebook collapse, especially when training samples are limited. To mitigate this, we introduce a finite scalar quantization method inspired by Mentzer et al. (2023), which enables learning a large motion vocabulary without requiring a lookup for corresponding tokens in the codebook for each entry. As a result, 2D-LFQ expands the motion codebook by at least two orders of magnitude, significantly boosting its representational capacity while maintaining efficiency.

We summarize our contributions as: **(1) MotionLib** — The first million-scale motion dataset, comprising over 1.2M motion sequences with hierarchical and detailed text annotations. **(2) MotionBook** — An effective motion encoding approach that combines a lossless and efficient motion feature with a lookup-free tokenizer, 2D-LFQ. Our 2D-LFQ preserves essential motion details, expands the motion encoder's capacity without token lookups, and improves the motion tokenizer's ability to leverage large-scale motion data. **(3) Being-M0** — A large motion model trained using MotionLib and MotionBook. We provide key insights into scaling both data and model size, identifying critical factors that influence the effectiveness of large motion models.

## 2. Related Work

**Large Language Models and Multi-Modality.** Substantial progress has been made in enhancing LLMs (Brown et al., 2020; Raffel et al., 2020; Chowdhery et al., 2022) with the ability to understand and respond to human commands, through a technique known as instruction tuning (Ouyang et al., 2022). Recent research has extended these capabili-

ties to the multimodal domain (Ye et al., 2023; Zheng et al., 2024; Feng et al., 2024; Mei et al., 2024), with notable work by Liu et al. (2023), who pioneered visual instruction tuning to create a highly adaptable assistant. In addition, Li et al. (2023a) integrated multimodal context into instruction data to further enhance performance. Subsequent studies (Zhang et al., 2023c) expanded this research by scaling up instructional data and incorporating image-rich text. Notably, Dai et al. (2023) developed InstructBLIP based on BLIP-2 (Li et al., 2023b), which features an advanced visual feature extraction mechanism to improve performance across vision-language tasks. Despite these, the application of multimodal models to human motion remains less competitive compared to current SoTA methods, although recent initiatives are beginning to explore this domain (Zhang et al., 2024d).

**Motion Vector Quantization.** Vector quantization (VQ) has been successful in generating high-quality images (Van Den Oord et al., 2017) and videos (Gupta et al., 2022; Yan et al., 2021). VQ-VAE first converts images into discrete representations and autoregressively models their distribution. Building on this, Lee et al. (2022) introduced residual quantization (RQ), which encodes images into a stacked map of discrete codes, efficiently reducing the spatial resolution of features. You et al. (2022) further developed hierarchical vector quantization (HQ), employing a pyramid scheme with two-level codes for image encoding. In motion generation, most existing approaches have adopted VQ or its variants to quantize human motions. However, the small codebook size in traditional VQ methods limits their ability to generalize and accurately represent the diversity of human motions. Although increasing the codebook size can improve representational capacity, it often leads to codebook collapse. Recently, Mentzer et al. (2023) demonstrated that discrete codes can be obtained via scalar quantization, where each scalar entry is quantized to the nearest integer through rounding. Similarly, Yu et al. (2023) introduced a lookup-free codebook that maps videos into compact discrete tokens, utilizing all codes without auxiliary losses and expanding the codebook size.

**Human Motion Generation** The task of motion generation involves creating human motion based on various inputs, such as text descriptions (Guo et al., 2022b), action labels (Cervantes et al., 2022) or motion prefixes (Liu et al., 2022). Among these, text-to-motion (T2M) generation has received the most attention due to the ease and flexibility of using natural language as input. Early approaches (Fragkiadaki et al., 2015; Ghosh et al., 2017) rely on deterministic motion modeling, which often produces averaged, blurry results. To overcome this, stochastic methods using models like GANs (Wang et al., 2020) or VAEs (Ali-akbarian et al., 2020) have been considered. For instance, T2M-GPT (Zhang et al., 2023a) extends the temporal VAE to capture the probabilistic relationship between text and

motion. Recently, Guo et al. (2024) proposed integrating residual quantization and masked modeling to improve traditional vector quantization (VQ). Lu et al. (2023) designed HumanTomato, a hierarchical VQ-VAE to separately encode body and hand motions. To better align with a motion auto-encoder, MotionCLIP (Tevet et al., 2022) incorporates CLIP (Radford et al., 2021) as the text encoder, bringing in more robust text priors. Additionally, Zhang et al. (2024d) and Jiang et al. (2023) explored the development of unified models based on LLMs that accept multimodal conditions (e.g., vision, text, and pose), enabling the generation of subsequent, preceding, or "in-between" motions. Despite leveraging the power of LLMs, these large motion models remain limited to in-domain text instructions and do not yet perform as competitively as existing SoTA methods.

In this work, we aim to bridge the gap between large language models and generalizable, reliable large motion models. To achieve this, We introduce MotionLib — the first million-level motion dataset designed to support extensive pretraining and comprehensive fair evaluation.

# 3. MotionLib: A Million-Level Motion Library

Data are the foundation of large motion models. With advancements in fields such as human pose detection, we are now able to extract high-quality motion sequences from vast amounts of online videos. Our MotionLib dataset contains over one million motion clips, totaling approximately 137 million frames. Each clip is annotated with fine-grained automatic pseudo-labels. A comparison with existing benchmarks is presented in Table 1. Examples of this dataset are shown in Figure 2. Our data curation pipeline involves the following main procedural steps. For more details, please refer to Appendix B due to limited space.

**Million-Level Motion Collection.** We begin by collecting more than 20 million videos from publicly available datasets (Kay et al., 2017) and online platforms such as YouTube. To ensure motion quality, we filter out videos that do not contain human figures. For each video, we use a pretrained model (Xu et al., 2022) to detect 2D human keypoints and filter out those without visible human activity. The human's bounding box is required to occupy a significant portion of the frame, making human movement clearly visible. Videos with only partially visible humans are removed to maintain the quality of the extracted motion. We adopt WHAM (Shin et al., 2024) to extract the SMPL parameters of collected videos, regressing 3D human motion in the world instead of the camera coordinate system.

**Hierarchical Motion Descriptions.** Existing benchmarks face inherent limitations in their text descriptions. Previous studies (Guo et al., 2022a) typically use a few short sentences to describe whole-body motions, neglecting finer

Table 1: Comparison with existing human motion datasets. More details can be found in Appendix B. In the table, B, H, and F refer to body, hand, and face, respectively. "hier" indicates that the text captions include hierarchical descriptions of motions, while "body" means the descriptions are not as detailed. "multi" and "single" specify whether the dataset contains multi-person scenarios or only single-person data. As the largest motion generation dataset and benchmark to date, MotionLib features at least 15× more motion and text data than previous datasets, along with additional modalities.

| | SEQ NUM | TEXT NUM | HOURS | MOTION | TEXT | RGB | DEPTH | BBOX | PERSON |
|---|---|---|---|---|---|---|---|---|---|
| KIT (Plappert et al., 2016) | 5.7K | 5.7K | 11.2 | B | body | ✗ | ✗ | ✗ | single |
| HumanML3D (Guo et al., 2022a) | 29.2K | 89K | 28.6 | B | body | ✗ | ✗ | ✗ | single |
| MotionX (Lin et al., 2024) | 81.1K | 142K | 144.2 | B,H,F | body | ✓ | ✗ | ✗ | single |
| MotionVerse (Zhang et al., 2024a) | 320k | 373k | - | B,H,F | body | ✓ | ✗ | ✗ | single |
| MotionLib | 1.21M | 2.48M | 1456.4 | B,H | hier | ✓ | ✓ | ✓ | single & multi |

details of individual body parts, such as the arms or legs. This restricts the model from performing more nuanced body comprehension and more flexible part-level motion control (e.g., raising only the left arm). Moreover, the richness of text labels often varies across different motions. For example, a large portion of the Motion-X dataset provides only a single sentence, and HumanML3D contains numerous similar text annotations. In contrast, MotionLib offers hierarchical textual annotations for each video inspired by Pi et al. (2023): (1) part-level description, where we carefully design a prompt format and use Gemini-1.5-pro (Reid et al., 2024) to generate detailed captions for individual body parts (e.g., left arm), assigning a dedicated sentence to each; (2) body-level description, which summarizes the whole body movement in a comprehensive paragraph containing 1–3 sentences. These hierarchical motion captions provide far more textual content and levels of detail than previously available. Their richness and structure are crucial for large language models to better understand and align with the motion modality.

**Motion and Description Refinement.** It is inevitable that the million-scale motions collected from diverse web sources initially contain noise. According to some previous studies (Holden, 2024), the low-quality motion may damage generation results. To enhance the quality of our collected motion, we first estimate precise 3D keypoints with another pre-trained model (Sárándi et al., 2023) to perform local-global motion optimization, following Lin et al. (2024). In addition, we train an RL-based policy $\pi_{refine}$ based on Luo et al. (2023) to elaborate and refine raw motions to obey physical laws (e.g., maintaining balance) and appear more realistic. $\pi_{refine}$ takes raw motion sequences as input, treating them as target poses, and generates new motion sequences that satisfy physical laws in a simulated environment, thus eliminating issues like jittering and foot-sliding. While effective, $\pi_{refine}$ may struggle with drastic movements in target poses, leading to slipping within the simulation. For such cases, we mark them with a smaller weight during pretraining, forcing our model to focus more on high-quality motion. To refine the description, since our text labels are

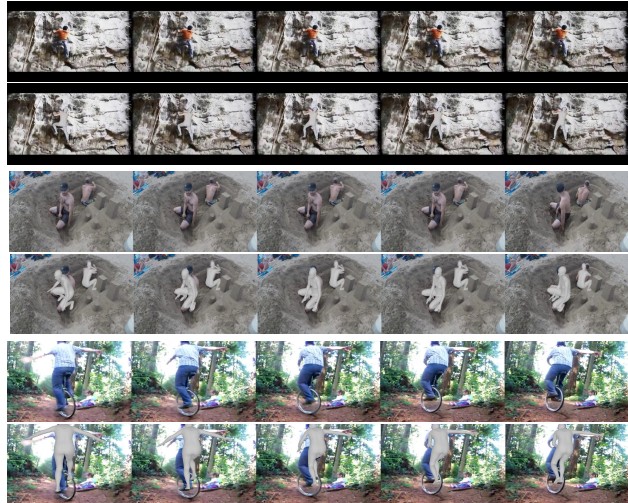

Figure 2: Examples from **MotionLib**, which encompasses a diverse range of human motions from web videos. It features various scenes, ranging from outdoor environments to indoor settings, and includes both clean, single-person scenarios as well as crowded, multi-person scenes. MotionLib provides over 2.4M motion-text pairs in total. The whole illustration with more examples can be seen in Figure 5.

structured into body and part levels, we condition GPT-4o with one level of text while using the other as the target to be refined. GPT-4o then assesses and refines the text content of the target level, enabling the generation of more precise and reliable descriptions.

## 4. Being-M0: Scaling up Large Motion Model

### 4.1. Overview

Inspired by the success of Large Language Models (LLMs) in multimodal tasks, we adapt a similar approach for motion generation, treating motion as a distinct "language" to be learned (Luo et al., 2020; Zhang et al., 2024c). Similar to previous LLM-based multimodal models, we treat motion as a foreign language. The overall framework is illustrated in Figure 3. Our large motion model, built

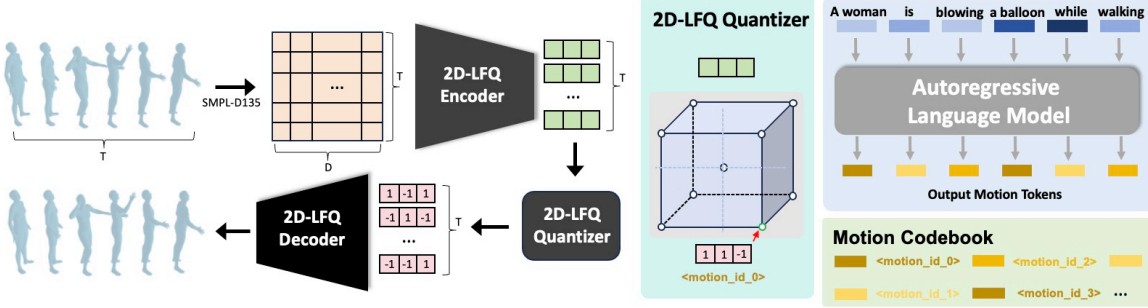

Figure 3: Overview of our large motion model named Being-M0, which can be divided into two stages. In the first stage (**left**), we pre-train a motion VQ-VAE to quantify motion sequences into tokens. In the second stage (**right**), we fine-tune an autoregressive language model to predict motion tokens.

on a pre-trained LLM, functions as a generative model that connects a motion tokenizer with the LLM backbone $\Theta$. The motion tokenizer encodes the features $\mathcal{M} = \{m_1, m_2, ..., m_T\}$ of raw motion sequence into token embeddings $\mathcal{V} = \{v_1, v_2, ..., v_n\} \in \mathbb{R}^{n \times d}$, where $n$ denotes the number of motion tokens and $d$ represents the dimensionality of each token. To integrate motion tokens into the LLM backbone, we incorporate $K$ discrete codes in the motion codebook as additional vocabulary for the LLM. Additionally, we introduce two special tokens, $<$mot$>$ and $</$mot$>$, to signify the start and end of motion sequences within the input/output streams. The LLM backbone $\Theta$ is built on a decoder-only architecture using causal transformers. The model generates outputs $\mathcal{Y} = \{y_1, y_2, ..., y_m\}$ in an auto-regressive manner, where $\mathcal{Y}$ corresponds to the generated motion sequence based on the provided motion-text input tokens. In this work, each motion-text pair in the MotionLib dataset is framed as an instruction-following instance $\{\mathcal{X}_Q, \mathcal{X}_M\}$, representing a question-answer interaction between the user and the motion model. The entire instructional dataset adheres to this unified format. To train our model, we optimize the negative log-likelihood over the predicted tokens which is defined as:

$$\mathcal{L}(\Theta) = -\sum_{j=1}^{L} \log P_\Theta(y_j | desc, \hat{y}_{1:j-1}), \quad (1)$$

where $\hat{y}$ and $y$ denote the input and target token sequences, respectively. $\Theta$ represents the model parameters, and $L$ is the length of the target sequence. The input description, $desc$, can be empty depending on the instruction provided.

**Two-Stage Training.** Similar to Liu et al. (2023), our training process incorporates motion-text alignment and motion instruction tuning. In the first stage, we utilize the entire MotionLib dataset to enable the model to learn fundamental motion-text correlations from diverse data. In the second stage, to better align with real-world human commands, we construct a set of over 250 instruction templates (e.g., "Show me a demonstration of

$<$Caption_Placeholder$>$ through movement."). We then select a subset of high-quality motions from MotionLib to formulate our instructional data. For each motion, we randomly choose three templates, inserting the corresponding text into the $<$Caption_Placeholder$>$. To further enhance naturalness and fluency, we leverage Gemini-Pro to refine these instructions, ensuring they align with human expression patterns. Finally, we construct an instruction fine-tuning dataset containing 900K instructions.

### 4.2. MotionBook: Towards Effective Motion Encoding

MotionBook consists of (1) a lossless feature representation for motion sequences $\mathcal{M}$, and (2) a 2D Lookup-Free Quantization (2D-LFQ) motion tokenizer to compress $\mathcal{M}$ into a series of discrete tokens.

**Lossless Motion Feature.** Conventional HumanML3D feature (H3D-Format) (Guo et al., 2022a) primarily focuses on joint positions and derives rotations using Inverse Kinematics (IK), which discards the original rotational information from SMPL features. Additionally, H3D-Format represents the root's angular velocity using a single scalar, capturing only Y-axis rotation, further limiting its expressiveness. Considering this, we adopt a feature named SMPL-D135 to represent motion $\mathcal{M}$. Each frame is encoded as $m \in \mathbb{R}^{135}$, structured as follows: (1) root (9D), 6D rotation $\mathbf{r}_{rot} \in \mathbb{R}^6$, 2D XZ-plane velocity $\mathbf{r}_{xz}^v \in \mathbb{R}^2$, and 1D height $r^y \in \mathbb{R}$. (2) body joints (126D), each of 21 key body joints are represented using 6D rotation vectors $\mathbf{j}^r \in \mathbb{R}^{21 \times 6}$. By directly encoding complete joint and root rotation data ($\mathbf{j}^r \in \mathbb{R}^{21 \times 6}$ and $\mathbf{r}_{rot} \in \mathbb{R}^6$), SMPL-D135 preserves critical motion information lost in H3D-Format while being more compact.

**2D Lookup-Free Motion Quantization.** Traditional motion tokenizer typically involves an encoder $\mathbb{E}$, a decoder $\mathbb{D}$, a codebook $\mathbb{C}$ and a quantizer $\mathbb{Q}$. It uses 1D embeddings to represent motion at each timestamp, which inevitably results in the loss of crucial information. Furthermore, this approach limits the tokenizer's ability to interpret part-level

Table 2: Comparisons under different model parameters and data sizes, showing their scaling law for motion generation.

| Decoder | #Inst. | #Param. | Motion-X-eval | | | MotionLib-eval | | |
|---|---|---|---|---|---|---|---|---|
| | | | R@1 ↑ | R@3 ↑ | FID ↓ | R@1 ↑ | R@3 ↑ | FID ↓ |
| Real | - | - | 0.514 | 0.831 | 0.046 | 0.297 | 0.634 | 0.004 |
| GPT-2 | 0.02M | 355M | 0.213 | 0.426 | 47.319 | 0.058 | 0.152 | 30.612 |
| GPT-2 | 0.08M | 355M | 0.468 | 0.792 | **0.083** | 0.114 | 0.281 | 22.077 |
| GPT-2 | 0.5M | 355M | 0.463 | 0.793 | 0.121 | 0.161 | 0.354 | 9.157 |
| GPT-2 | 1.2M | 355M | 0.472 | 0.791 | 0.112 | 0.166 | 0.375 | 6.936 |
| LLaMA-2 | 0.02M | 7B | 0.216 | 0.433 | 47.538 | 0.059 | 0.158 | 29.643 |
| LLaMA-2 | 0.08M | 7B | 0.472 | 0.798 | 0.166 | 0.118 | 0.294 | 21.593 |
| LLaMA-2 | 0.5M | 7B | 0.468 | 0.799 | 0.178 | 0.164 | 0.369 | 9.146 |
| LLaMA-2 | 1.2M | 7B | 0.475 | 0.798 | 0.156 | 0.171 | 0.380 | 6.632 |
| LLaMA-3 | 0.02M | 8B | 0.216 | 0.435 | 47.906 | 0.059 | 0.162 | 29.257 |
| LLaMA-3 | 0.08M | 8B | 0.483 | 0.815 | 0.122 | 0.120 | 0.301 | 21.295 |
| LLaMA-3 | 0.5M | 8B | 0.483 | 0.817 | 0.113 | 0.166 | 0.368 | 8.973 |
| LLaMA-3 | 1.2M | 8B | 0.486 | 0.820 | 0.117 | 0.173 | 0.386 | **6.029** |
| LLaMA-2 | 0.02M | 13B | 0.223 | 0.446 | 47.210 | 0.061 | 0.169 | 29.143 |
| LLaMA-2 | 0.08M | 13B | 0.488 | 0.820 | 0.156 | 0.124 | 0.314 | 21.001 |
| LLaMA-2 | 0.5M | 13B | 0.490 | 0.819 | 0.145 | 0.174 | 0.374 | 8.824 |
| LLaMA-2 | 1.2M | 13B | **0.491** | **0.823** | 0.133 | **0.185** | **0.391** | 6.221 |

motions. To tackle these, we treat the motion sequence $\mathcal{M} = \{m_1, m_2, ..., m_T\}$ as a single-channel image, representing each motion sequence as $\mathcal{M} \in \mathbb{R}^{T \times D \times 1}$. Each motion feature $m_i$ is divided into $P$ components, capturing distinct features of motion, such as root orientation, joint rotation, and foot contact. Our motion encoder then converts $\mathcal{M}$ into a feature map $\mathbb{E}(\mathcal{M}) \in \mathbb{R}^{\lfloor T/\alpha \rfloor \times P \times d}$, where $\alpha$ denotes the temporal downsampling ratio. This approach ensures that each body part is tokenized separately, allowing for more granular, part-level motion encoding and decoding.

In addition, previous motion tokenizers are restricted from capturing the full diversity of human motion due to their small codebook sizes. A common approach to address this limitation is to expand the vocabulary. However, excessively enlarging the codebook can result in codebook collapse, where only a small subset of tokens in the codebook is used, offering minimal improvements even potentially degrading the model's performance. To enable effective learning with larger codebooks suited for million-scale datasets like MotionLib, we adopt a lookup-free quantization strategy. A more effective way is to reduce the dimensionality of code embeddings (Mentzer et al., 2023), which limits the representational capacity of individual tokens and encourages more efficient learning across a larger vocabulary. Similar to Yu et al. (2023), we reduce the embedding dimension of the codebook to zero by replacing the codebook $\mathbb{C} \in \mathcal{R}^{K \times d}$ with an integer set $\mathbb{C}$ with $|\mathbb{C}| = K$. Here, $\mathbb{C}$ is the Cartesian product of single-dimensional variables $\mathbb{C} = \bigtimes_{i=1}^{d} C_i$, where $C_i = \{-1, 1\}$ and $d$ is equal to $\log_2 K$. Given a feature vector $z \in \mathbb{R}^d$, our motion quantizer $Q(\cdot)$ converts each dimension of the quantized representation into:

$$Q(z_i) = \arg\min_{c_{ik}} ||z_i - c_{ik}|| = -\mathbb{1}\{z_i \le 0\} + \mathbb{1}\{z_i > 0\},$$

$$(2)$$

where $c_{ij}$ denotes the $j$-th value of $C_i$. The token index is computed as $Index(z) = \sum_{i=1}^{d} 2^{i-1} \mathbb{1}\{z_i > 0\}$. To train the tokenizer, we employ a standard combination of reconstruction, perceptual, and commitment losses, along with an entropy penalty to promote better codebook utilization (Yu et al., 2023). Importantly, we exclude the use of GAN loss, as it is found to negatively impact training stability.

## 5. Experiments

Due to space limitation, we present more details about the metrics and model implementation in Appendix A, as well as additional experiments in Appendix C.

### 5.1. Experimental Setup

**Datasets.** Our investigation is first conducted on HumanML3D (Guo et al., 2022a) and Motion-X (Lin et al., 2024). HumanML3D comprises 14,616 motion sequences from the AMASS dataset (Mahmood et al., 2019), paired with 44,970 textual descriptions. Motion-X, a more recent dataset, includes approximately 81,000 motion sequences. To validate our conclusions on larger-scale data, we also carry out experiments on our MotionLib dataset with two variants: MotionLib-0.5 and MotionLib-full. MotionLib-0.5 contains 500K clips, while MotionLib-full encompasses all 1.2M clips of our collected data. Following standard practice, each dataset is split into training, validation, and test sets in proportions of 85%, 5%, and 15%, respectively.

**Evaluation.** For the motion generation task, we employ the evaluation metrics following (Guo et al., 2022a): Motion-retrieval Precision (R-Precision), Multimodal Distance (MMDist), and Frechet Inception Distance (FID). R-Precision assesses the consistency or matching between

generated motions and input text descriptions. MMDist measures the average distance between generated motion and corresponding text in feature space. FID evaluates the realism and quality of generated motions by comparing the distribution of generated motion features with that of real motion features. In addition, we evaluate our motion tokenizer on the motion reconstruction task. Besides FID, this task is also measured by MPJPE, which measures the average distance between predicted and ground truth joint positions across all joints, in millimeters.

### 5.2. Discussion of Scaling up Motion Generation

In this section, we investigate the impact of model size and data scale for motion generation. For evaluating the models, we use the same motion autoencoder architecture (Guo et al., 2022a) retrained separately on Motion-X and MotionLib. This model is used to evaluate the performance of motion generation models on their respective test sets. We categorize training data into four scales: 0.02M (HumanML3D only), 0.08M (Motion-X only), 0.5M (MotionLib-0.5), and 1.2M (MotionLib-full). For fair comparison, we employ the same motion tokenizer, maintaining consistency across experiments to validate our conclusions.

**Does increasing model size benefit motion generation?**
Yes. As shown in Table 2, the experimental results demonstrate that increasing model size leads to a stable performance enhancement when the training data is kept constant. For example, our model achieves best R@1 (0.491) on Motion-X when using LLaMA2-13b, which is 0.016 higher than LLaMA2-7b's 0.475. On the larger Motion-Lib dataset, the best R@1 of LLaMA2-13b reaches 0.185, which is 0.014 higher than LLaMA2-7b's 0.171. This series of results strongly validates the trend of performance gains scaling with model capacity. Consequently, larger models show a greater ability to capture the diverse and intricate patterns and relationships within human motion.

**Does increasing the data scale benefit motion generation?**
Yes. As illustrated in Table 2, when using the same foundation model, increasing the scale of training data leads to substantial improvement on MotionLib testing set, aligning with our expected scaling laws. When using LLaMA2-13b as the LLM backbone for our model, training with 1.2M data achieves an R@1 of 0.185, which is 0.011 and 0.061 higher than performance using 0.5M data and 0.08M Motion-X, respectively. This improvement is particularly pronounced in the R-precision metric, emphasizing the critical role of data scale in enhancing semantic alignment between generated motions and text prompts. It's worth to note that the performance of models trained on Motion-X also improves, after pretraining on MotionLib. This demonstrates that large-scale pretraining allows the model to have a better initialization for motion-language alignment.

Table 3: Comparison with existing SoTA methods on HumanML3D. Results marked with * represent values reproduced using the official code, while unmarked results are taken from the original papers. [1] and [2] denote different works with the same model name. For fair comparison, experiments here are conducted using HM3D-Format feature.

| | Decoder | R@1↑ | R@3↑ | FID↓ | MMDist↓ |
|---|---|---|---|---|---|
| Real | - | 0.511 | 0.797 | 0.002 | 2.974 |
| MLD | - | 0.481 | 0.772 | 0.473 | 3.196 |
| MotionDiffuse | - | 0.491 | 0.782 | 0.630 | 3.113 |
| ReMoDiffuse | - | 0.510 | 0.795 | 0.103 | 2.974 |
| Fg-T2M++ | - | 0.513 | 0.801 | 0.089 | 2.925 |
| LMM | - | 0.525 | 0.811 | **0.040** | 2.943 |
| T2M-GPT | GPT-2 | 0.492 | 0.775 | 0.141 | 3.121 |
| DiverseMotion | GPT-2 | 0.510 | 0.802 | 0.072 | 2.941 |
| MoMask | - | 0.521 | 0.807 | 0.045 | 2.958 |
| MotionGPT[1,*] | T5 | 0.409 | 0.667 | 0.162 | 3.992 |
| MotionGPT[1] | T5 | 0.492 | 0.778 | 0.232 | 3.096 |
| MotionGPT[2,*] | LLaMA2-13B | 0.367 | 0.654 | 0.571 | 3.981 |
| MotionGPT[2,*] | LLaMA-13B | 0.363 | 0.633 | 0.592 | 4.029 |
| MotionGPT[2] | LLaMA-13B | 0.411 | 0.696 | 0.542 | 3.584 |
| MotionLLM | Gemma-2b | 0.482 | 0.770 | 0.491 | 3.138 |
| AvatarGPT | LLaMA-13B | 0.389 | 0.623 | 0.567 | - |
| MotionGPT-v2 | LLaMA3.1-8B | 0.496 | 0.782 | 0.191 | 3.080 |
| Being-M0-VQ | LLaMA2-13B | 0.519 | 0.803 | 0.166 | 2.964 |
| Being-M0-LFQ | LLaMA2-13B | **0.528** | **0.820** | 0.141 | **2.875** |

**Does our large motion model outperform SoTA models?** Yes. We evaluate our model Being-M0 on the widely adopted HumanML3D benchmark. We compare its performance against a variety of SoTA approaches. This includes diffusion-based methods such as MLD (Chen et al., 2023), MotionDiffuse (Zhang et al., 2022), ReMoDiffuse (Zhang et al., 2023b), Fg-T2M++ (Wang et al., 2025) and LMM (Zhang et al., 2024a). It also includes autoregressive models like T2M-GPT (Zhang et al., 2023a), DiverseMotion (Lou et al., 2023), and MoMask (Guo et al., 2024). We also compare against LLM fine-tuning methods like MotionGPT (Jiang et al., 2023; Zhang et al., 2024d), MotionGPT-v2 (Wang et al., 2024), MotionLLM (Wu et al., 2024), and AvatarGPT (Zhou et al., 2024). As shown in Table 3, our model, which utilizes LLaMA2-13B as the motion decoder, achieves SOTA R-Precision performance. Existing T2M methods can be categorized intow two types: (1) specialist models optimized on specific datasets, and (2) LLM-based generalist models (aiming for broader instruction and task generalization via LLMs), like AvatarGPT, MotionGPT-v2 and our Puppet. Puppet excels among generalist models and remains highly competitive on R@1, R@3, and MMDist compared to specialist models. This highlights our model's ability to generate motion sequences that are better aligned with text descriptions and of higher quality.

**Does large motion model excel in out-of-distribution setup?** Yes. We present the results in Table 4. This ablation is essential for further validating the true generalization

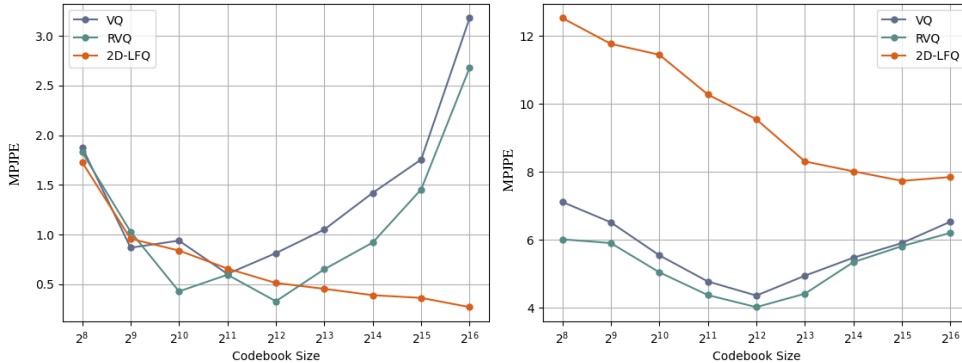

Figure 4: Comparison with different motion quantization on Motion-X (**left**) and MotionLib (**right**). We only show MPJPE (↓) results here due to space limitation, with FID results shown in Figure 12.

Table 4: Ablation of out-of-domain evaluation on UNSEEN-90K dataset, where $\#N$ denotes we use $N$ subsets of MotionLib for training.

| TRAIN SET | R@1 ↑ | R@3 ↑ | FID ↓ |
|---|---|---|---|
| Real | 0.176 | 0.379 | 0.076 |
| HumanML3D | 0.034 | 0.112 | 82.674 |
| MotionX | 0.051 | 0.141 | 70.547 |
| MotionLib-#11 | 0.098 | 0.218 | 11.930 |

Table 5: Ablation results of motion instruction tuning.

| TRAIN SET | R@1 ↑ | R@3 ↑ | FID ↓ |
|---|---|---|---|
| Real | 0.514 | 0.831 | 0.046 |
| Pretrain | 0.471 | 0.788 | 0.103 |
| Instruction tuning | 0.488 | 0.821 | 0.093 |

Table 6: Ablation results of hierarchical description vs. single-level description.

| TRAIN TEXT | R@1 ↑ | R@3 ↑ | FID ↓ |
|---|---|---|---|
| Real | 0.297 | 0.634 | 0.004 |
| single-level | 0.162 | 0.371 | 7.018 |
| hierarchical | 0.166 | 0.375 | 6.936 |

Table 7: Ablation results of different motion features. Here, "FPS" denotes the speed to recover original motion information (e.g., rotation).

| Motion Feat | R@1 ↑ | R@3 ↑ | FPS ↑ |
|---|---|---|---|
| H3D-D263 | 0.514 | 0.831 | 0.41 |
| SMPL-D130 | 0.517 | 0.851 | >100 |
| SMPL-D135 | 0.529 | 0.850 | >100 |
| SMPL-D263 | 0.521 | 0.843 | >100 |
| SMPL-D268 | 0.523 | 0.855 | >100 |

capabilities of large motion models, as the improvements observed in Table 2 may result from the inclusion of additional in-domain data similar to testbeds. In this setup, we select 11 subsets from MotionLib, totaling 90K samples (UNSEEN-90K), for evaluation. The remaining subsets are used for training. These 11 subsets include both synthetic and real-world data. They cover domains like fitness, person-object interaction, social activities, and diverse environments such as labs and outdoors. The training set, on the other hand, primarily consists of Motion-X and web-derived data. This ensures that the test set is entirely composed of out-of-domain (OOD) samples. We compare the performance of models trained on HumanML3D, Motion-X, and Motion-#11, all utilizing the GPT-2 architecture, where $\#N$ denotes the number of training subsets. The results on the OOD test set clearly demonstrate that the model trained on MotionLib significantly outperforms those trained on HumanML3D and Motion-X, particularly in terms of R@1 and R@3 metrics. These findings strongly highlight the superior generalization ability of large motion models when handling unseen OOD data, especially when trained on diverse, large-scale datasets. This demonstrates the value of utilizing a large amount of web-based motion data.

**Does motion instruction tuning help?** Yes. We compare experiments with and without instruction tuning on Motion-X. As shown in Table 5, instruction tuning data effectively improves various metrics, with R@1 increased by 0.017

and FID increased by 0.010. This indicates a benefit for the model's further language-motion alignment capability. Instruction tuning also makes the model more user-friendly, proving it a worthwhile optimization.

**Does richer text description help?** Yes. To investigate the effectiveness of richer text descriptions, we compared the training results using hierarchical descriptions, which include both body-level and part-level labels, against those using only whole-body description. As shown in Table 6, training with hierarchical descriptions improves both R@1

Table 8: Ablation results of different motion tokenizer trained on HumanML3D on the motion reconstruction task.

| Tokenizer | #Num. | #Param. | HumanML3D | | Motion-X | | MotionLib | |
|---|---|---|---|---|---|---|---|---|
| | | | FID ↓ | MPJPE ↓ | FID ↓ | MPJPE ↓ | FID ↓ | MPJPE ↓ |
| VQ-VAE | 512 | 19.43M | 0.078 | 69.2 | 0.852 | 106.4 | 5.324 | 123.6 |
| H$^2$VQ | 512 | - | - | - | - | 62.34 | - | - |
| RQ-VAE | 512 | 19.43M | **0.052** | **37.5** | 0.568 | 56.9 | 4.026 | 78.2 |
| 2D-LFQ | 16384 | 108.35M | 0.092 | 45.6 | **0.295** | **54.1** | **2.315** | **64.1** |

and R@3 by 0.004 compared to single-level descriptions. These results indicate that hierarchical text significantly enhances the model's semantic understanding, leading to improved semantic matching of generated motions.

### 5.3. Effectiveness of Motionbook

**Motion Feature Design.** We conduct an ablation study to evaluate the impact of different motion feature design, including H3D-Format, SMPL-D130, SMPL-D135, SMPL-D263, and SMPL-D268 (detailed in Appendix A.3). As shown in Table 7, retrieval performance remains consistent across all features, with no significant differences. Similarly, quantitative evaluations of rendered samples reveal no noticeable variation among different features. This indicates that none of the tested features provide a clear performance advantage. However, we observe that H3D-Format discards some original motion information, as its rotations are obtained through inverse kinematics. This requires additional time-consuming processing to restore missing details — an issue that complicates downstream applications. Considering simplicity, computation cost, and information completeness, we finally select SMPL-D135 as the motion feature.

**2D-LFQ Motion Tokenization.** We compare our proposed 2D-LFQ against three quantization approaches: vanilla vector quantization (VQ), residual vector quantization (RVQ), and H$^2$VQ (Lu et al., 2023)[1], across various codebook sizes ranging from $2^8$ to $2^{16}$. The number of parameters for RVQ/VQ and 2D-LFQ are 19.43M and 108.35M, respectively. To evaluate the generalization ability of each alternative, all results are trained on HumanML3D dataset. As shown in Table 8, 2D-LFQ outperforms its counterparts on out-of-domain and larger datasets like Motion-X and MotionLib. In addition, 2D-LFQ continues to enhance performance as the codebook size increases as illustrated in Figure 4, while RVQ and VQ experience diminishing returns or performance degradation with larger codebooks. Our deeper analysis attributes these gains to better codebook utilization by 2D-LFQ. Figure 11 (RIGHT) illustrates that the utilization rates for VQ and RVQ begin to decline once the codebook size exceeds $2^{10}$, which corresponds to the peak performance for these methods, whereas the utilization

of 2D-LFQ continues to increase with larger codebooks.

## 6. Conclusion

In this paper, we explore how to advance the field of large motion model. To address the issue of data scarcity in this domain, we introduce MotionLib, the first million-level dataset comprising over 1.2 million motions with hierarchical texts. Building on MotionLib, we present key insights into scaling up both data and model size for large-scale motion training. Furthermore, we propose MotionBook, a novel motion encoding approach designed to maximize the benefits when trained on extensive motion data. Motion-Book incorporates compact yet lossless features to represent motion, and introduces a novel 2D-LFQ motion quantization method that treats each motion sequence as a 2D image, constructing a finite-scale codebook that eliminates the need for token lookups. Leveraging these advancements, we train Being-M0, a large motion model that achieves SoTA results compared to current counterparts.

### Acknowledgements

This work was supported by NSFC under Grant 62450001. The authors would like to thank the anonymous reviewers for their valuable comments and advice.

### Impact Statement

This paper introduces MotionLib, the first million-scale motion dataset, and a large motion model named Being-M0 trained upon it. This work significantly advances the field of large motion models and lays a foundation for future research. It aims to improve a model's understanding and generation of human motion, positively impacting areas like gaming and robotics. In terms of ethical considerations, all motion data comes from open-source and online videos, we will carefully review examples before release to avoid any potential ethical limitations such as copyright issue. We acknowledge the positive implications of such technological advancements but also recognize potential risks, like misuse for misleading content. We are committed to responsible development and will continuously monitor the broader societal impact of this technology.

---

[1]Only reported number is shown (MPJPE on Motion-X), since no official code or successful re-implementation has been found.

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

# Appendix

**Roadmap.**    In this appendix, we first provide the additional details of evaluation, model implementation and motion feature design in Section A. Then, we introduce additional details of our dataset MotionLib and its construction process in Section B. Finally, we carry out a thorough experiments and responding analysis in Section C.

## Table of Contents

# A. Details of Evaluation and Implementation

## A.1. Evaluation Metrics

We employ the following metrics to assess the quality and alignment of generated motions: (1) Frechet Inception Distance (FID): This metric evaluates the overall motion quality by comparing the distributional differences between high-level features of generated and real motions. (2) Motion-retrieval Precision (R-Precision) and Multimodal Distance (MMDist): These metrics measure the semantic alignment between textual inputs and generated motions. R-Precision evaluates retrieval accuracy at top-1, top-2 and top-3 levels, while MMDist quantifies the distance between matched text and motion pairs. To validate the effectiveness of our motion tokenizer, we also perform experiments on the motion reconstruction task. This task is measured using both Mean Per Joint Position Error (MPJPE) and FID, where MPJPE computes the average distance (in millimeters) between predicted and ground-truth joint positions across all skeleton joints.

## A.2. Implementation Details

For a fair comparison with previous works, we implement oue model Being-M0 based on two varionts of motion tokenizers: one with a vector quantized (VQ) codebook and another with a 2D lookup-free (2D-LFQ) codebook. By default, our Being-M0 is trained using 2D-LFQ. For the motion tokenizer, we implement the VQ codebook $\mathbb{C} \in \mathbb{R}^{1024 \times 512}$ with an embedding dimensionality of $d = 512$. The resulting discrete codes are incorporated as additional vocabulary for the LLM. As a comparison, the LFQ codebook has a size of $2^{16} = 16384$. The motion encoder $\mathbb{E}$ uses a temporal downsampling rate of $\alpha = 4$. We experiment with four large language model (LLM) architectures to construct our large motion model: GPT2-medium (Radford et al., 2019), LLaMA2-7b, LLaMA2-13b (Touvron et al., 2023), and LLaMA3.1-8b (Dubey et al., 2024). The motion tokenizer is trained with a learning rate of 1e-4 and a batch size of 256 for 300K iterations. For training the large motion model, full parameter tuning is performed on $8 \times$A800 GPUs with a batch size of 1024 over 100 epochs. The learning rate is set to 2e-4 for GPT2-medium and 2e-5 for the LLaMA models.

## A.3. Design of Different Motion Features

In this section, we provide a detailed introduction of diverse motion feature formats used in our experiments to highlight their key differences:

- **H3D-Format:** This format is proposed by Guo et al. (2022a) and is widely used by most recent motion generation works. H3D-Format includes relative joint positions (63 dimensions), relative 6D rotations of key joints (126 dimensions), joint velocities (66 dimensions), and foot contact information (4 dimensions). Rotation information is derived from position data using Inverse Kinematics (IK), which may result in the loss of original rotational details. The root node parameters consist of 4 dimensions: 1 for r-rotation (angular velocity), 2 for xz-velocity, and 1 for y-height.

- **SMPL-D130:** This format uses relative 6D rotations of key joints (126 dimensions) and root node parameters (4 dimensions). The root node parameters include 1 dimension for r-rotation, 2 for xz-velocity, and 1 for y-height.

- **SMPL-D263:** Building on SMPL-D130, this format adds redundant position features (derived from forward kinematics of the SMPL model) and 4 dimensions of foot contact information. The root node parameters remain the same as in SMPL-D130.

- **SMPL-D135:** This format employs relative 6D rotations of key joints (126 dimensions) and 9-dimensional root node parameters. The root node parameters include 6 dimensions for 6D root rotation, 2 for xz-velocity, and 1 for y-height.

- **SMPL-D268:** Extending SMPL-D135, this format incorporates redundant position features (identical to SMPL-D263) and foot contact information. The root node parameters are consistent with those in SMPL-D135.

# B. Details of Dataset — MotionLib

In this section, we provide a detailed introduction of how **MotionLib**, the million-level motion generation dataset, is constructed, with more insights into its characteristics and additional data examples shown in Figure 5.

## B.1. Motion Data Collection

We begin by elaborating on the process of extracting raw motion data from web videos.

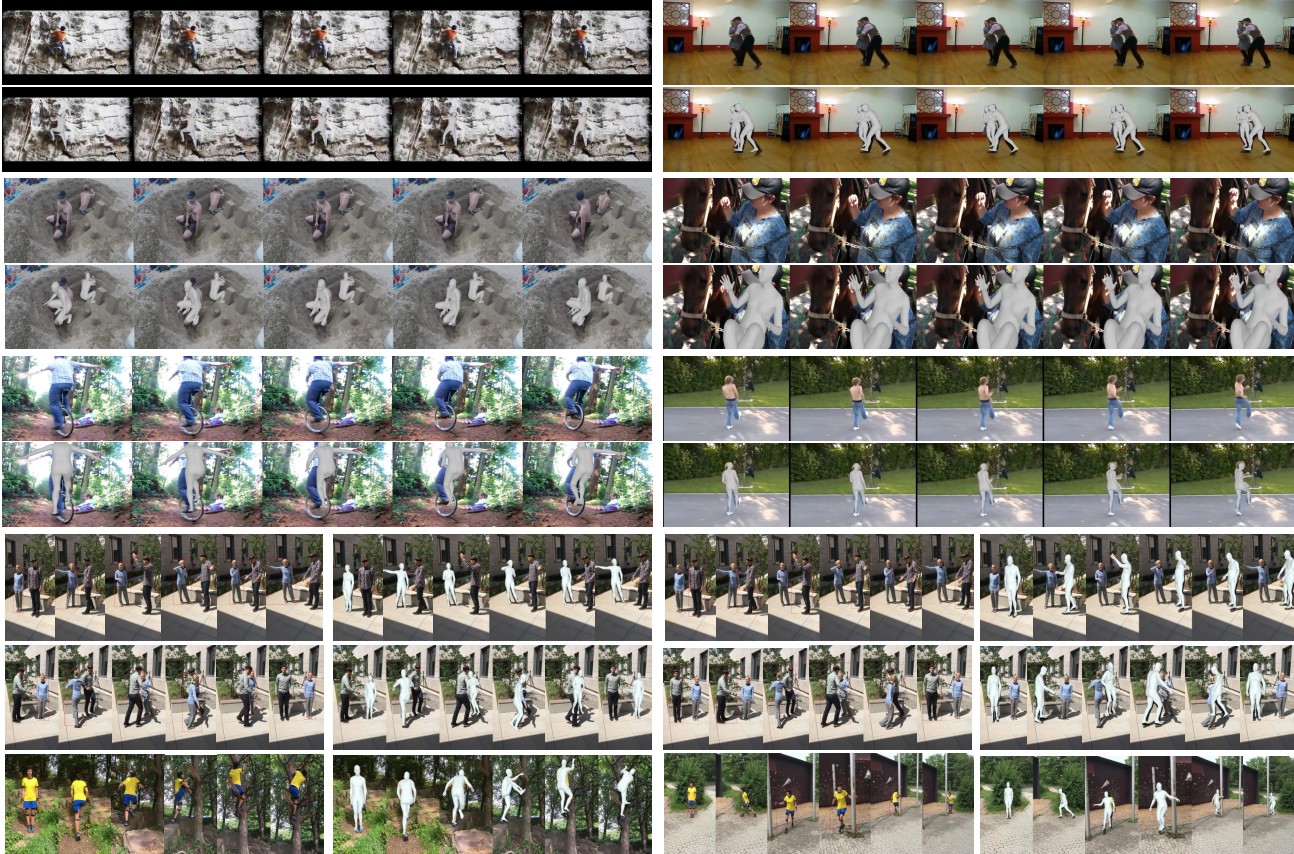

Figure 5: Illustration of examples in MotionLib, each sample is a motion sequnce extracted from an online video.

**Short Boundary Detection.** We observe that many segments in these videos lack human presence, necessitating the identification and extraction of portions relevant to human motion. To achieve this, we follow a structured step-by-step procedure: For videos shorter than 30 seconds or those with clearly defined temporal boundaries, we either use the entire clip directly or segment it based on the provided boundaries. For videos longer than 30 seconds, we first apply a scene detection model to divide the video into coarsely segmented parts. Each segment is then refined into shorter clips through the following steps: (1) At the start of a segment, we identify the human with the largest bounding box as the anchor and track their trajectory. (2) If the trajectory is interrupted, the start and end times of the interruption define a new clip boundary. (3) The process repeats by selecting the next largest visible human in subsequent frames and tracking their trajectory. (4) This continues until no humans remain visible in the segment. (5) Clips without visible humans are discarded. If a resulting clip exceeds 60 seconds, we randomly divide it into sub-clips, ensuring each is shorter than one minute.

**Occlusion and Blur Filtering.** Occlusion and motion blur are common challenges in human-related videos. To mitigate these issues, We first sample key frames from each clip and apply a pretrained 2D keypoint detector to extract skeleton keypoints for each human. If a significant portion of the keypoints has confidence scores below a predefined threshold, the motion is considered occluded and excluded from further processing. Additionally, we leverage a visual foundation model, such as Segment Anything (Kirillov et al., 2023), to generate segmentation masks for each frame. If a large object is detected obstructing the human, the corresponding motion data is filtered out. To address motion blur, we track the trajectory of each human whose motion data is being extracted. For timestamps where keypoint confidence scores are low, we smooth the trajectory using adjacent detection results, ensuring continuity and accuracy.

### B.2. Motion Data Refinement

To ensure physically plausible motion data, we refine the extracted motion using an RL-based policy (Luo et al., 2023). Before this refinement step, we enhance motion quality following the methodology of Lin et al. (2024). We begin by

estimating 2D human keypoints and their confidence scores using the pretrained VitPose model (Xu et al., 2022), then infer 3D keypoints via another pretrained 3D keypoint estimation model (Sárándi et al., 2023). To improve stability and consistency, we apply temporal smoothing and enforce 3D bone length constraints during triangulation. As previously mentioned, we fit the SMPL-X body model (Pavlakos et al., 2019) to each frame in MotionLib using the approach from Shin et al. (2024), followed by iterative optimization to align model parameters with the estimated 2D and 3D joint positions. We further refine global motion and camera poses using a global motion optimization technique based on Yuan et al. (2022), ensuring consistency with the original video evidence. For motion sequences affected by noise or occlusion, we employ RoHM (Zhang et al., 2024b) to reconstruct complete and physically plausible motions.

## B.3. Motion Description Generation

In this paper, we employ Gemini-1.5-Pro (Reid et al., 2024) as a large multimodal model (LMM) to generate text labels for motion data. For each video clip, we first crop and track the human body using the corresponding bounding boxes. The LMMs are then tasked with analyzing the human's physical movements and their spatial positions within the global space to produce detailed motion descriptions. A key distinction from previous datasets is the granularity of these descriptions. Instead of simply generating an overall description of the human's movements, we prompt the LMMs to focus on specific body parts by dividing the body into upper-body and lower-body sections, which enables the generation of part-specific descriptions (referred to as "part-level" descriptions). Figure 6 illustrates the corresponding used prompt. Additionally, a comprehensive summary of the whole body's movements (referred to as "body-level" descriptions) is also included.

Table 9: Text quality evaluation of different datasets. We use both "text-only" and "visual-text align" to score the text description quality. Here, the score is produced by GPT-4o.

| Eval Strategy | HumanML3D | MotionX | MotionLib |
|---|---|---|---|
| Text-only | 1.386 | 1.703 | **3.837 (+2.134)** |
| Visual-text align | 3.081 | 2.252 | **3.823 (+0.742)** |

## B.4. Motion Description Quality Assessment

To ensure the quality of text descriptions, we propose two evaluation strategies. Table 9 compares the results of our MotionLib with HumanML3D and Motion-X.

**Text-only Evaluation.** We first conduct an automated evaluation of motion descriptions in MotionLib using GPT-4o-mini (OpenAI, 2024). Unlike the Gemini model used for text generation, we employ a different LMM (GPT-4o) as the evaluator to mitigate model bias. Each motion description is rated on a 1-to-5 scale based on the following criteria:

- 1 point (very poor): Vague, irrelevant to the motion, or contains severe grammatical errors.

- 2 points (poor): Lacks detail, specificity, or contains clear inaccuracies.

- 3 points (fair): Broadly reflects the motion but lacks depth and may have minor errors.

- 4 points (good): Accurate, detailed, and clearly conveys the motion process.

- 5 points (excellent): Precise, comprehensive, and fluent, providing in-depth motion analysis.

ollowing the same methodology, we evaluate descriptions from HumanML3D and Motion-X. As shown in Table 9, MotionLib achieves an average score of 3.837, significantly outperforming Motion-X (1.386) and HumanML3D (1.703). These results underscore the superior quality of MotionLib's motion descriptions.

**Visual-text Alignment Evaluation.** Simply relying on text-based input to generate a score may not be ideal, as LLMs are prone to hallucination when they lack visual guidance to contextualize the content. To address this, we pair text descriptions with their corresponding rendered motion videos, and feed them into the GPT-4o for scoring. Here, we extract key frames from the rendered video, concatenating them into a larger image as the input of GPT-4o, considering GPT-4o does not accept videos as input. GPT-4o will evaluate the alignment between motion descriptions and rendered visual content, and output a

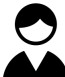

Begin by providing a general overview of the person's current action (e.g., walking, sitting, interacting) within the BBOX area. Then, proceed with a detailed breakdown, focusing exclusively on the physical movements and positions of the person within the BBOX. For the upper body, describe the position and movement of the arms, hands, shoulders, and torso. For the lower body, detail the position and movement of the legs, feet, and overall balance. Ensure the description strictly covers physical actions without mentioning facial expressions, clothing, or environmental elements outside the BBOX.

**Example:**

The person is standing still, observing something in front of them.

- **Upper body：** Their arms hang relaxed by their sides, with the shoulders slightly back and the chest open. The torso is upright, with minimal movement, indicating a calm, neutral stance.
- **Lower body：** Both feet are planted firmly on the ground, shoulder-width apart. The knees are slightly bent, and their weight is evenly distributed between both legs.

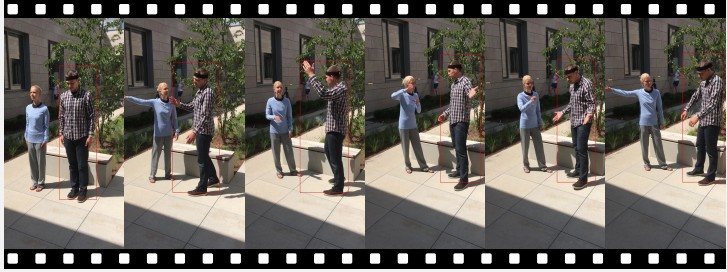

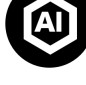

The person is standing within the designated area, engaging in a conversation seemingly directed toward someone positioned off-camera to the left.

**Upper Body:**

* **Arms:** Initially held loosely at the sides, the arms transition to various positions throughout the interaction. At times, they rise to chest level with palms open, suggesting an explanatory gesture. Occasionally, one or both arms extend outwards, indicating direction or emphasis.
* **Hands:** Hand movements correspond with arm gestures. Palms face upwards and outwards during open-handed motions, then relax to a neutral position when the arms are at rest.
* **Shoulders:** Shoulders remain relatively relaxed throughout, with subtle shifts in position reflecting the arm movements. They don't appear tense or raised, implying a generally comfortable stance.
* **Torso:** The torso largely remains stationary, facing forward, with slight turns coinciding with the shifting weight distribution of the lower body.

**Lower Body:**

* **Legs:** Legs maintain a comfortable stance, slightly apart, with the weight appearing balanced. There's a subtle shift in weight distribution as they adjust their stance.
* **Feet:** Feet remain planted on the ground, primarily shoulder-width apart. The positioning suggests a grounded and stable stance.
* **Overall Balance:** The individual appears balanced and at ease throughout the interaction, with movements suggesting engagement in the conversation rather than discomfort or restlessness.

Figure 6: The Prompt template to generate part-level motion description in videos based on powerful large multimodal models (LMMs), such as Gemini-1.5-pro and GPT-4o-mini. For each sample in MotionLib, we provide "body-level" (UP) and "part-level" (DOWN) labels to distinguish between whole-body and partial motion descriptions.

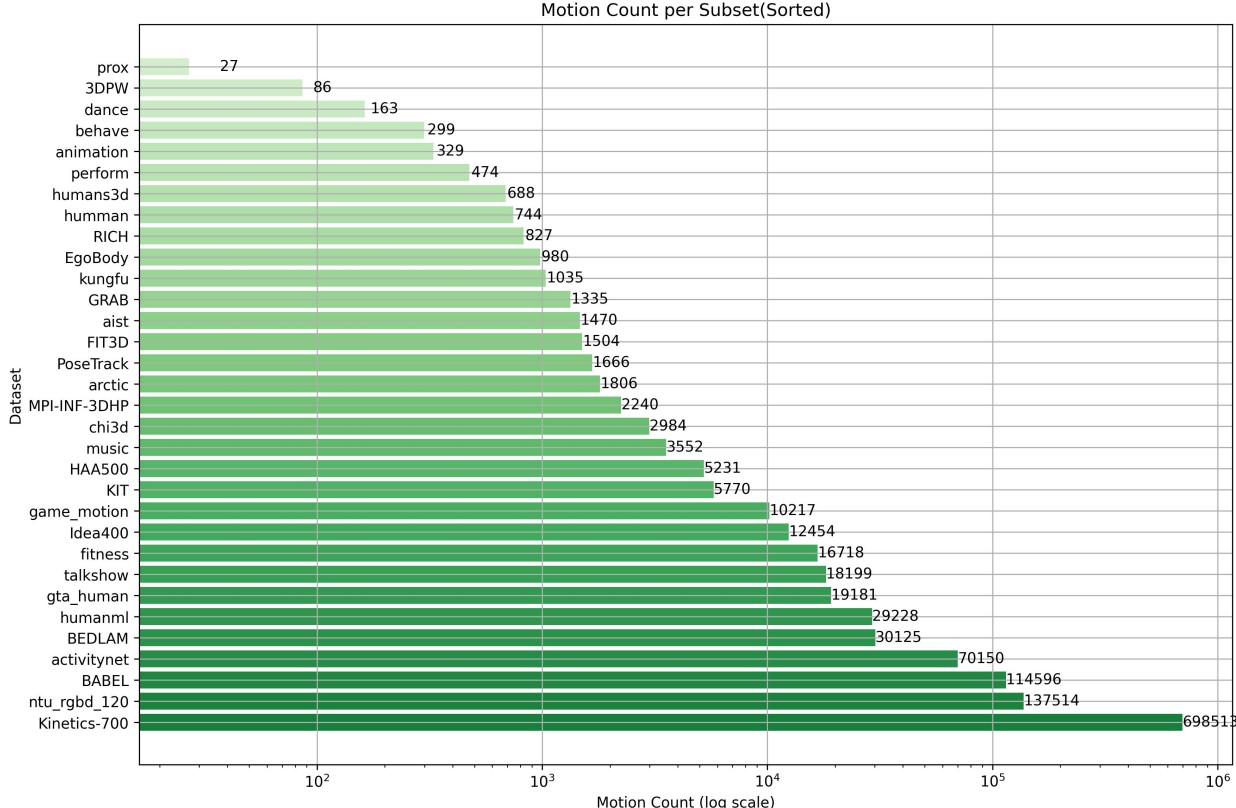

Figure 7: The data scale distribution of motion sequences for different subsets in MotionLib.

score following the same criteria above. As a result, our MotionLib achieves an average score of 3.823, while Motion-X and HumanML3D score 2.252 and 3.081, respectively, again confirming the quality advantage of MotionLib's text descriptions.

### B.5. Statistic Analysis of Data and Word Distribution

**Data Distribution.** MotionLib comprises over 1.2 million motion sequences collected from various public datasets and web videos. A significant portion of MotionLib is derived from open-source, human-related datasets, such as 698.5K motions from Kinetics-700 (Kay et al., 2017) and 137.5K motions from NTU-RGBD-120 (Lin et al., 2024). Additionally, MotionLib integrates motions from other established datasets, including BEDLAM and GTA-Human. MotionLib also includes subsets of the Motion-X collection, covering a diverse range of categories such as Animation, Perform, Dance, AIST, Kungfu, GRAB (Taheri et al., 2020), Music, Idea400, HAA500 (Chung et al., 2021), Game Motion, and Fitness. It is worth noting that the Motion-X subsets constitute only a small portion of the overall MotionLib dataset (around 6.7%). Figure 7 illustrates the scale distribution of motion sequences within the subsets of MotionLib. We also count the average frame number of each subset, as shown in Figure 8. Given the high cost of collecting and annotating video data, we also recognize the untapped potential of images for motion understanding. To explore this, we collected approximately 600K images and extracted human poses, which were repeated across 64 frames and treated as motion sequences. **While using static data for dynamic motion generation remains a controversial topic, this static data is not included in the 1.2 million motion sequences of MotionLib in the main paper and is not claimed as part of our primary contributions.** Nevertheless, we conduct experiments with this static data and hope it will inspire future research into the potential of static data for dynamic motion understanding, as can be seen in Apppendix C.3.

**Word Distribution.** To further explore the annotated motion text, we compute the word cloud from the entire text corpus in MotionLib. The word cloud of body-level and part-level descriptions can be seen in Figure 9 and Figure 10, respectively. In Figure 9, we observe that the body-level texts primarily highlight the high-level human activities, such as standing, sitting, and walking. In contrast, Figure 10 illustrates that part-level descriptions focus more on specific body-part movements, such the torso, shoulders, legs, and arms. We believe that this hierarchical structure of text corpus can enhance the alignment

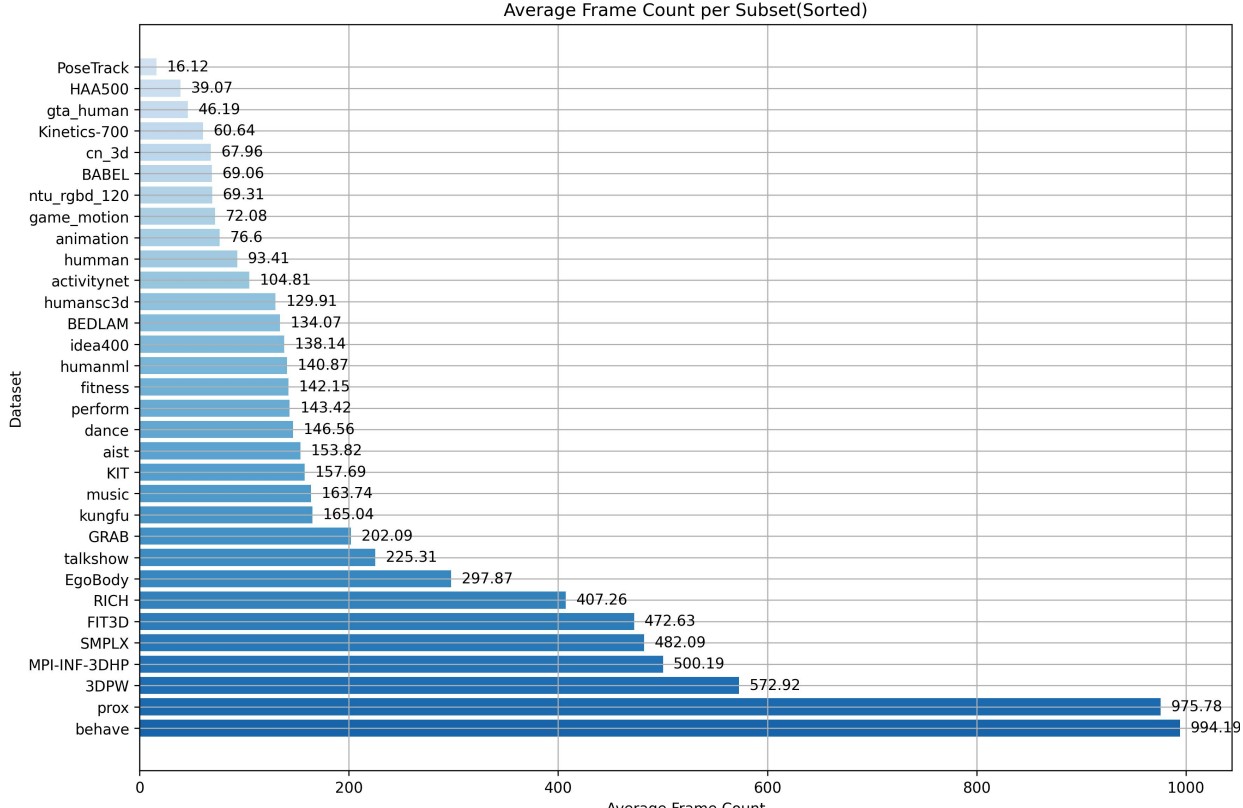

Figure 8: The length distribution across different subsets in MotionLib

between LLM and motion modality, therefore improving the understanding of motion.

## C. Additional Experimental Analysis

In this section, we present additional experimental analysis that could not be included in the main paper due to space constraints.

### C.1. Ablation of Large Motion Model Training

Without further notification, the following experiments are carried out on MotionLib test set. For simplicity, we employ Vector Quantization (VQ) for motion encoding in the following ablation experiments.

#### C.1.1. LoRA vs. Full Parameter Fine-tuning

We conduct an ablation study comparing LoRA and full parameter fine-tuning. As illustrated in Table 10, LoRA fine-tuning struggles to achieve competitive results. We hypothesize that this limitation stems from the introduction of new motion tokens, which require significant parameter updates for the large language model to effectively learn and interpret these additional tokens. The restricted capacity of LoRA fine-tuning appears inadequate to meet these demands, highlighting the challenges of adapting to such changes with limited parameter updates.

#### C.1.2. Learning from Scratch vs. Fine-tuning

We compare the performance of fine-tuning GPT-2 against training it from scratch with random initialization on Motion-X-eval. As demonstrated in Table 11, fine-tuned models consistently achieve superior results compared to those trained from scratch, particularly when trained on the HumanML3D dataset and evaluated on Motion-X. The significant performance gap underscores the importance of text pre-training, which enhances the model's ability to comprehend text descriptions

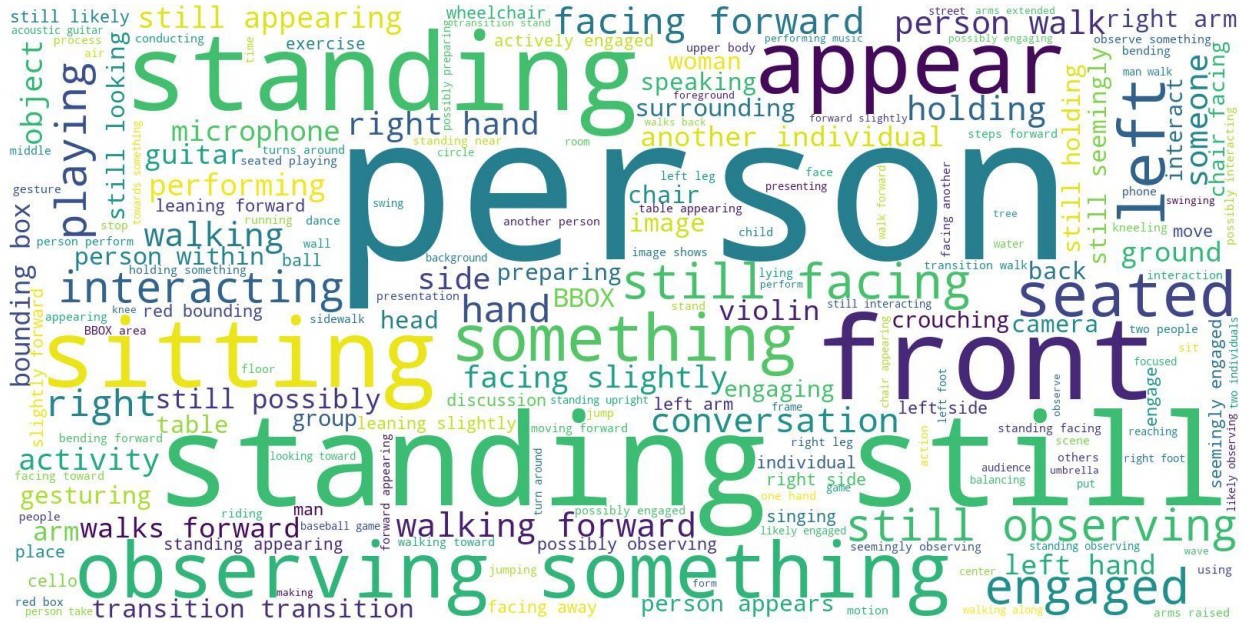

Figure 9: Word cloud of body-level motion descriptions in MotionLib.

Table 10: Ablation results of LoRA tuning vs. full-parameter fine-tuning.

| TRAIN TYPE | R@1 ↑ | R@3 ↑ | FID ↓ | MMDist ↓ |
|---|---|---|---|---|
| Real | 0.297 | 0.634 | 0.004 | 2.068 |
| LoRA | 0.157 | 0.354 | 9.287 | 4.832 |
| full-param | 0.166 | 0.375 | 6.936 | 4.484 |

and improves its generalization capabilities across diverse tasks. This observation also validates the effectiveness of our hierarchical text labels. With richer textual content, full-parameter fine-tuning enables better alignment between the motion modality and the language model, further leveraging the strengths of pre-trained motion representations.

Table 11: Ablation results of learning from scratch vs. fine-tuning.

| #Inst | From Scratch | R@1 ↑ | R@3 ↑ | FID ↓ | MMDist ↓ |
|---|---|---|---|---|---|
| Real | - | 0.514 | 0.831 | 0.046 | 2.438 |
| 0.02M | Yes | 0.042 | 0.116 | 17.932 | 8.957 |
| 0.02M | No | 0.213 | 0.426 | 47.319 | 7.666 |
| 0.08M | Yes | 0.461 | 0.784 | 0.116 | 2.862 |
| 0.08M | No | 0.468 | 0.792 | 0.083 | 2.798 |

### C.1.3. WITH VS. WITHOUT DESCRIPTION MASK

We also investigate the impact of different mask strategies of input sequence on model performance on Humanml3d: Specifically, we compare two strategies: training with and without masking input motion description. As shown in Table 12, our results indicate that the second strategy yields better performance. This improvement over the first approach can be attributed to the strategy's ability to prevent catastrophic forgetting of text understanding, ensuring that the model retains its capacity to interpret textual inputs. Additionally, it helps reduce overfitting to motion patterns in the training data, thereby

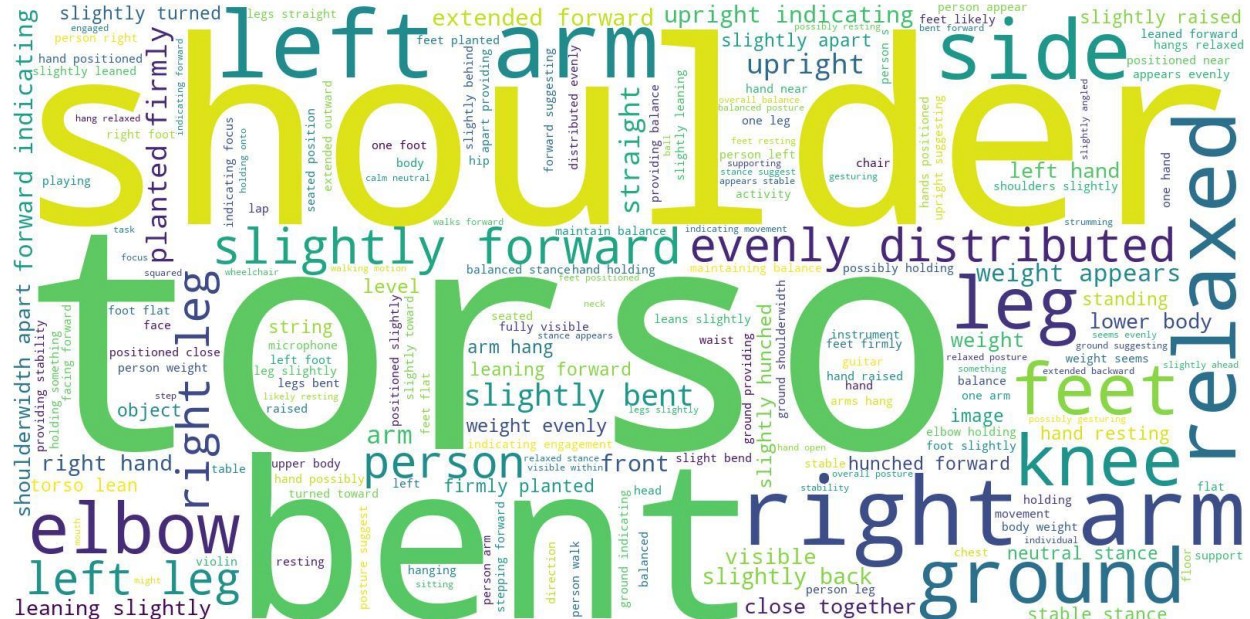

Figure 10: Word cloud of part-level motion descriptions in MotionLib.

enhancing the model's generalization capabilities.

Table 12: Ablation results of different mask strategy of input sequence.

| Mask Strategy | R@1 ↑ | R@3 ↑ | FID ↓ | MMDist ↓ |
|---|---|---|---|---|
| Real | 0.511 | 0.797 | 0.002 | 2.974 |
| with description mask | 0.388 | 0.650 | 0.680 | 3.919 |
| w/o description mask | 0.466 | 0.752 | 0.101 | 3.234 |

### C.1.4. ENCODER-DECODER VS. DECODER-ONLY

Existing large motion models typically adopt either an encoder-decoder or an decoder-only architecture (Jiang et al., 2023). To evaluate their performance, we carry out abaltions by training an encoder-decoder model, T2M-GPT (Zhang et al., 2023a) on the MotionLib dataset and comparing it with a decoder-only model based on GPT-2 medium. As shown in Table 13, despite having comparable parameter counts, T2M-GPT struggles to produce competitive results. This limitation can be attributed to the inherent constraints of text encoding capabilities by using "CLIP+random-initialized decoder", which hinder the model's ability to comprehend a broader spectrum of motion-related language. In contrast, we find that large motion models based on decoder-only LLMs, which jointly train text tokens and motion tokens, achieve superior text-motion semantic alignment and exhibit stronger motion generation capabilities.

Table 13: Ablation results of Encoder-Decoder vs. Decoder-only architecture.

| Arch | Model Name | #Param. | R@1 ↑ | R@3 ↑ | FID ↓ | MMDist ↓ |
|---|---|---|---|---|---|---|
| - | Real | - | 0.297 | 0.634 | 0.004 | 2.068 |
| enc-dec | T2M-GPT | 380M | 0.161 | 0.364 | 7.085 | 4.773 |
| dec-only | GPT-2 Medium | 355M | 0.166 | 0.375 | 6.936 | 4.484 |

### C.1.5. SLOW CONVERGENCE OF LARGE MOTION MODELS

To evaluate the convergence speed of large motion models, we train GPT-2, LLaMA2-7B, and LLaMA3-8B for 300 epochs on Motion-X. The training curves, measured by R@1 performance, are illustrated in Figure 11 **LEFT**. We observe that all large motion models nearly converge by 200 epochs, with larger models exhibiting faster convergence rates. Initializing these models with pre-trained weights significantly accelerates convergence. However, compared to large multimodal models like LLaVA (Liu et al., 2023), large motion models require more epochs to capture the intricate representations of motion sequences. We attribute this slower convergence to the limited representation capacity of the motion tokenizer, which currently supports only 1024 motion tokens. This limitation highlights the need to optimize the motion tokenizer and expand its representation space. To address this, we explore the 2D-LFQ quantization method as a promising alternative.

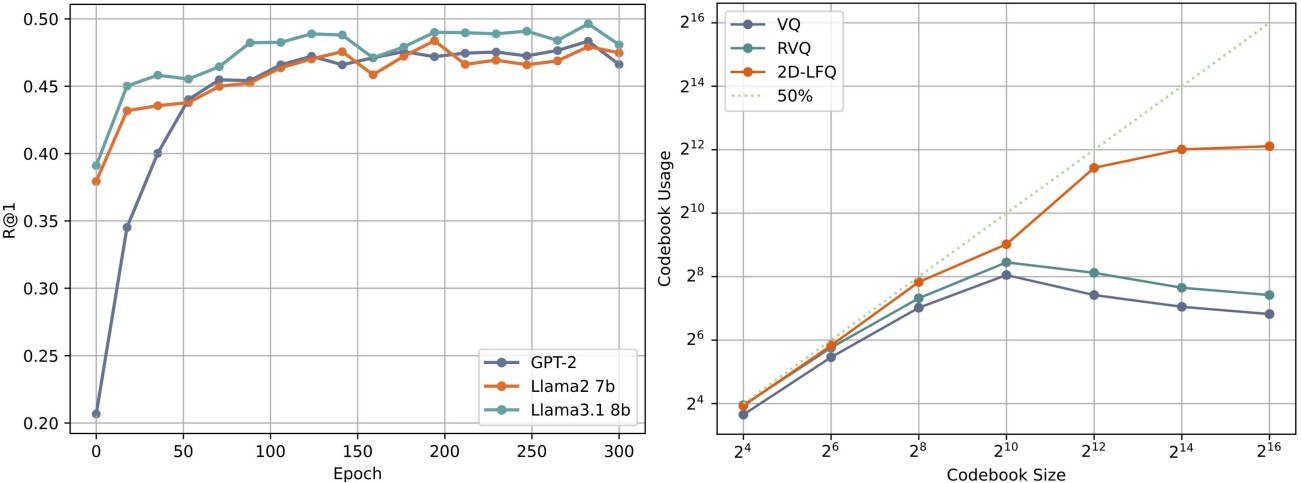

Figure 11: **LEFT**: Training curves with Y-axis denoting R@1 retrieval accuracy. All these models are trained for 300 epochs at most and are evaluated every 1000 steps; **RIGHT**: Ablation of codebook usage of different quantization methods

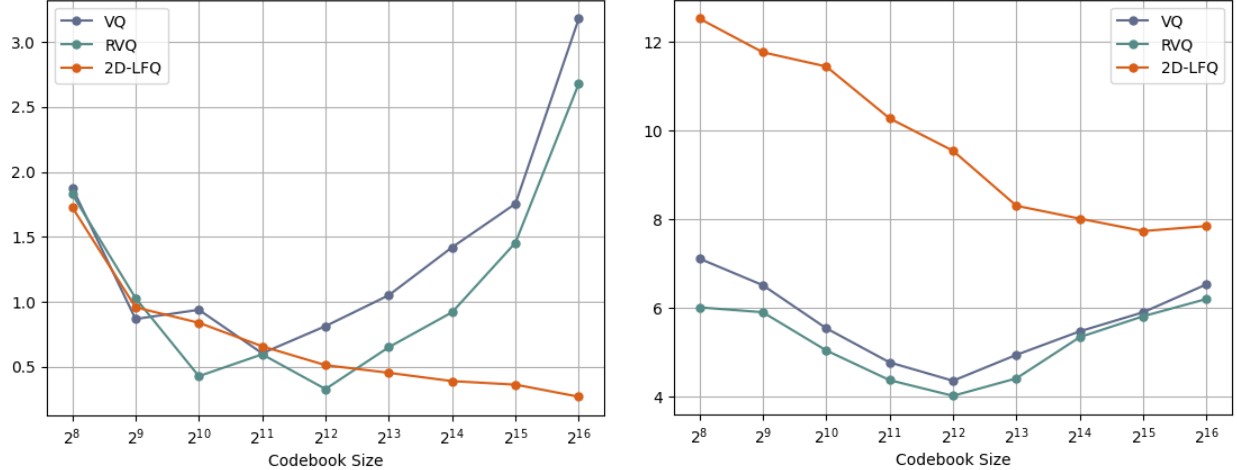

Figure 12: Comparison with different motion quantization on the Motion-X (**LEFT**) and MotionLib dataset (**RIGHT**). The Y-axis denotes FID (↓).

## C.2. Additional Experiments of Motion Quantization

### C.2.1. ADDITIONAL FID RESULTS ON MOTION-X

First, we provide additional FID results on Motion-X in Figure 12. It is worth noting that while our motion quantizer performs worse than RQ-VAE on the smaller HumanML3D dataset, it surpasses both VQ and RQ when evaluated on the

larger Motion-X and MotionLib benchmarks, as can be seen in Table 8. This suggests that our approach offers a greater advantage when applied to larger datasets, highlighting its improved generalization compared to previous methods.

### C.2.2. ABLATION OF 2D-LFQ VS. 1D-LFQ

To validate the effectiveness of our 2D strategy for motion quantization, we compare the 2D-LFQ method with its 1D counterpart (which is functionally equivalent to VQ except for the quantization strategy). The results, presented in Table 14, show that 2D quantization in LFQ significantly outperforms the 1D version. This underscores the superior ability of 2D quantization to enhance the representational capacity of the motion tokenizer.

Table 14: Ablation of 2D motion quantization vs. its 1D version.

|  |  |  | HumanML3D | | Motion-X | | MotionLib | |
| --- | --- | --- | --- | --- | --- | --- | --- | --- |
| Tokenizer | #Num. | #Param. | FID↓ | MPJPE↓ | FID | MPJPE | FID | MPJPE |
| 1D-LFQ | 16384 | 19.43M | 3.85 | 52.5 | 2.783 | 78.9 | 10.358 | 80.1 |
| 2D-LFQ | 16384 | 108.35M | **1.769** | **45.6** | **0.295** | **54.1** | **7.853** | **64.1** |

### C.3. Discussion of Static and Synthetic Data.

Although images only capture static poses, exploring their effectiveness for motion generation remains valuable. Additionally, synthetic data may play a significant role, as both image and synthetic data are far more accessible than dynamic videos. With this in mind, we collect approximately 600K static human-related images and extract their corresponding human poses. Each pose is repeated 60 times to create a 60-frame sequence. During pretraining, we introduce specific language prompts, such as "keep the action still", to explicitly guide the model in distinguishing between static and dynamic actions. Such prompt-based method effectively differentiates between different motion distributions. To validate the effectiveness of synthetic and static data, we conduct a series of ablation experiments, as shown in Table 15. We train GPT-2 medium on three data configurations: MotionLib without synthetic data, the 1M-scale MotionLib dataset, and MotionLib augmented with static data. The model is trained for 300 epochs with a learning rate of 2e-4 and evaluated on two benchmarks: a subset of our collected static and synthetic data, and the MotionLib testing set. Our results indicate that incorporating both static data and synthetic data leads to slight improvements in R-Precision. However, given the marginal gains, further exploration is necessary to fully realize the potential of these data types. We emphasize that static data is excluded from MotionLib in our main paper, and all experiments outside this section are conducted exclusively on dynamic motion data.

Table 15: Ablation of the effectiveness of static and synthetic data.

|  | Static-Sync-eval | | | | MotionLib-eval | | | |
| --- | --- | --- | --- | --- | --- | --- | --- | --- |
| TRAIN SET | R@1↑ | R@3↑ | FID↓ | MMDist↓ | R@1↑ | R@3↑ | FID↓ | MMDist↓ |
| Real | 0.290 | 0.563 | 0.011 | 3.480 | 0.196 | 0.474 | 0.006 | 1.647 |
| MotionLib with syn data | 0.111 | 0.248 | 57.719 | 8.412 | 0.167 | 0.396 | 1.740 | 2.323 |
| MotionLib | 0.120 | 0.252 | 55.983 | 8.175 | 0.166 | 0.393 | 1.780 | 2.356 |
| MotionLib + static data | **0.264** | **0.542** | **0.516** | **4.007** | **0.168** | **0.399** | **1.614** | **2.300** |

### C.4. Discussion of Current Evaluation Metric's Limitation

During our experiments, we notice the evaluation metrics are not as robust as we expect. Considering this, we carry out experiments for further analysis. Table 16 presents results using different retrieval models as evaluators. Here, we employ the same dual-encoder architecture following Guo et al. (2022a) as the retrieval model, but trained it on two distinct datasets: HumanML3D and Motion-X, where HumanML3D is a subset of Motion-X. Unlike robust visual models such as CLIP (Radford et al., 2021), these retrieval models are constrained by their small parameter size and limited training data. As shown in the table, performance using these two evaluators is much worse than our results in the main paper using evaluator trained on MotionLib. More importantly, when using the model trained on the larger Motion-X dataset, performance on HumanML3D decrease. This suggests that training on the broader Motion-X dataset negatively impacts

| Text Prompt | Generated Motion Sequences |
|---|---|

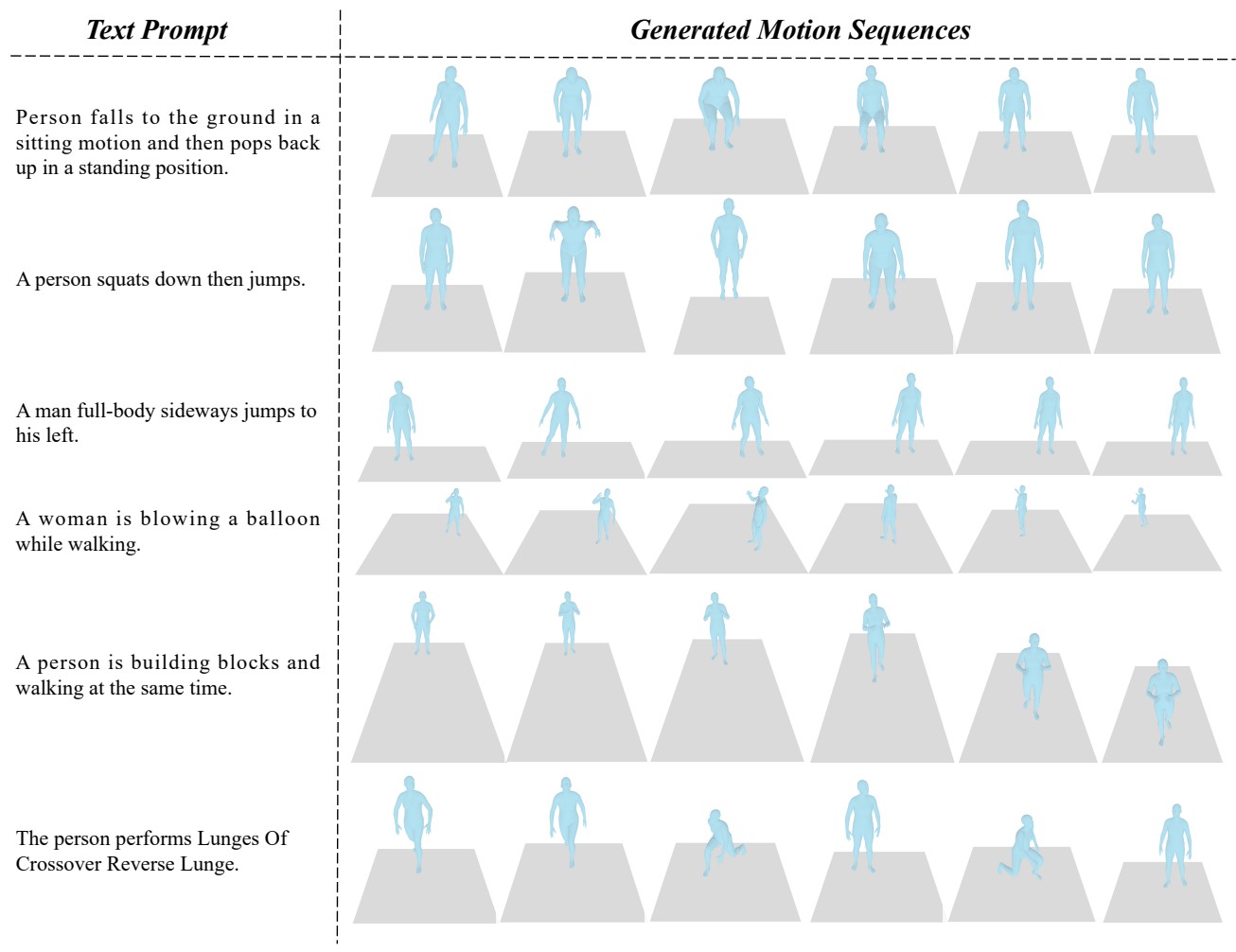

| | | | | |
|---|---|---|---|---|
| Person falls to the ground in a sitting motion and then pops back up in a standing position. | | | | |
| A person squats down then jumps. | | | | |
| A man full-body sideways jumps to his left. | | | | |
| A woman is blowing a balloon while walking. | | | | |
| A person is building blocks and walking at the same time. | | | | |
| The person performs Lunges Of Crossover Reverse Lunge. | | | | |

Figure 13: Quantitative examples of motions generated by our large motion model Being-M0.

Table 16: Comparison of evaluations using different retrieval models on the HumanML3D test set.

| Decoder | #Inst. | #Param. | Humanml3d-eval | | | Motion-X-eval | | |
|---|---|---|---|---|---|---|---|---|
| | | | R@1↑ | R@3↑ | FID↓ | R@1↑ | R@3↑ | FID↓ |
| Real | - | - | 0.511 | 0.797 | 0.002 | 0.496 | 0.821 | 0.038 |
| GPT-2 | 0.02M | 355M | 0.466 | 0.752 | **0.101** | 0.358 | 0.651 | **0.050** |
| GPT-2 | 0.08M | 355M | 0.462 | 0.744 | 0.208 | 0.362 | 0.656 | 0.754 |
| LLaMA-2 | 0.02M | 7B | 0.497 | 0.778 | 0.214 | 0.378 | 0.671 | 0.122 |
| LLaMA-2 | 0.08M | 7B | 0.474 | 0.758 | 0.452 | 0.376 | 0.673 | 0.518 |
| LLaMA-3 | 0.02M | 8B | 0.500 | 0.783 | 0.173 | 0.380 | 0.675 | 0.094 |
| LLaMA-3 | 0.08M | 8B | 0.499 | 0.786 | 0.264 | 0.393 | 0.696 | 0.591 |
| LLaMA-2 | 0.02M | 13B | **0.519** | **0.803** | 0.166 | 0.395 | 0.695 | 0.105 |
| LLaMA-2 | 0.08M | 13B | 0.504 | 0.790 | 0.393 | **0.400** | **0.700** | 0.637 |

R-Precision performance on the HumanML3D subset, even though HumanML3D is part of Motion-X. We hypothesize the unexpected results arise from the limited generalization capability of the current evaluator. FID, a standard metric for generation tasks, is typically computed using a pretrained evaluator. In image generation, FID relies on robust visual

encoders, such as InceptionNet (Szegedy et al., 2015), trained on millions of images. In contrast, the evaluator used for motion generation is a lightweight motion autoencoder with a small parameter scale (Guo et al., 2022a), trained on only limited data (e.g., HumanML3D with 20K motions). Such small-scale training data may hinder its ability to generalize effectively, leading to unreliable semantic alignment between text and motion, particularly for unseen motions. In fact, some recent works (Petrovich et al., 2023; Voas et al., 2023) have begun to recognize these limitations. For instance, Petrovich et al. (2023) proposed a simple yet effective approach for text-to-3D human motion retrieval, while Voas et al. (2023) highlighted that existing metrics are sensitive to embedding space quality and often misalign with human perception. These findings emphasize the need for more robust and fair evaluation metrics for large motion models. We believe addressing this challenge is of significant value and plan to explore it further in future work.

## C.5. Additional Qualitative Results

We present additional examples to visualize the human motions generated by our large motion model, **Being-M0**, trained on the **MotionLib** dataset, as shown in Figure 13. The results demonstrate that our model can produce motion sequences that closely align with the input texts, highlighting the effectiveness of MotionLib as a training resource.

