# OpenReview forum: "Scaling Large Motion Models with Million-Level Human Motions"
_ICML.cc/2025/Conference — ICML 2025 poster_

### Official Review · Reviewer_q9qB · 2025-03-12

**Overall Recommendation:** 3

**Summary:**

The paper introduces MotionLib, the first million-level dataset for motion generation, which is 15× larger than previous datasets and includes hierarchical text descriptions. Using MotionLib, the authors train Puppet, a large-scale motion model that demonstrates robust generalization across diverse human activities, including unseen scenarios. To improve motion encoding, they propose Motionbook, which includes:
- A lossless feature representation for motion data.
- A novel 2D lookup-free motion tokenizer that balances fine-grained details and expanded codebook capacity.

Their study emphasizes the importance of scaling both data and model size for advancing motion generation and highlights key insights for achieving generalist motion models. Experimental results show existing models struggle with out-of-domain generalization, whereas MotionLib enables better scalability, positioning it as a benchmark comparable to ImageNet for motion data.

**Claims And Evidence:**

Overall, the explanations are clear and well-structured. They effectively convey the key ideas and provide a solid understanding of the topic.

**Essential References Not Discussed:**

None.

**Experimental Designs Or Analyses:**

The experimental design appears sound overall.

**Methods And Evaluation Criteria:**

The evaluation criteria sounds reasonable and appropriate for assessing the proposed method.

**Other Comments Or Suggestions:**

### Post-Comments of Rebuttal
After discussing with other reviewers, I find that the concerns regarding motion quality still persist. Therefore, I will adjust my score accordingly. That said, I am inclined to lean toward acceptance due to the contribution of the enlarged dataset. I hope the authors will continue refining this dataset post-submission, unlike Motion-X, which saw limited adoption due to its subpar quality.

**Other Strengths And Weaknesses:**

### Strength
- The dataset is significantly larger compared to previous datasets and provides more fine-grained text descriptions. The authors claim that they optimize motion quality using reinforcement learning (RL).
- Experiments validate the effectiveness of scaling and propose a lossless motion representation that is better suited for recovery.
- The 2D encoder indeed enhances performance by preserving more fine-grained information.

### Weakness
- The supplementary material provides too few visualizations, making it difficult to fully assess the richness and quality of the dataset. Some videos are noticeably blurry, which raises concerns about data quality. I also observed that the motion quality is not very high—even after RL optimization, there are still cases of character deformation. It’s unclear whether this is due to perspective issues from Matplotlib rendering or an inherent flaw in the model itself, but either way, it’s a problem that shouldn’t be ignored.
- The claimed motion representation improvement is questionable. From the supplementary videos, foot sliding and pose jittering are still quite prevalent, making it hard to see the claimed benefits. If the proposed motion representation is supposed to enhance stability, where is the concrete proof of its effectiveness? The visual evidence so far doesn’t convincingly support this claim.

**Questions For Authors:**

Please see weakness.

**Relation To Broader Scientific Literature:**

Overall, this work significantly contributes to scaling motion generation and refining motion representation learning. By introducing MotionLib and Motionbook, it bridges the gap between small-scale motion datasets and large-scale generalist models, setting a new benchmark for future research in human motion synthesis and multimodal learning

**Theoretical Claims:**

No theoretical claims in this paper.

---

> ### Author Rebuttal · Authors · 2025-03-30
>
> Dear Reviewer,
>
> Thank you for your thoughtful review and positive feedback. We have carefully considered your questions and suggestions and provide our responses below. Please let us know if you require further clarification.
>
> ---
> ## **Response to: Insufficient visualizations in supplementary material, video blurring, motion quality, and deformation issues.**
> - **Regarding visualization quantity and video blurring**:
>
>     Thank you for your feedback. Due to conference supplementary material size constraints (typically 100MB), we were limited in the number of visual examples we could include and had to compress video resolution. To better demonstrate MotionLib’s richness, we plan to release a dedicated website with additional high-resolution examples upon final publication.
>
> - **Regarding motion quality and deformation**:
>
>     We acknowledge that motion data extracted and refined from web videos may still exhibit imperfections (e.g., deformations), even after RL optimization. However, we also feel more positive since our motion model is significantly enhanced using such motion data, demonstrating the scaling law in this task (Table 2, 4). This suggests the larger potential of our MotionLib dataset. As we keep refining this dataset using additional strategies, the gain of performance and generalization could be more significant. Considering this, we consider the building of MotionLib as a long-term iterative process, and our future work can incorporate stricter filtering or advanced algorithms, in addition to current steps we adopted (e.g., 3D keypoint optimization, physics constraints, and RL strategies; see Appendix B.2).
>
>   In addition, we suggest the issue of data quality and quantity is a trade-off for large-scale pretraining. Motivated by LLM's pretraining, we believe combining large-scale pre-training with motion fine-tuning on high-quality subsets can further mitigate quality issues while preserving scalability benefits.
>
> ## **Response to: Questioning the effectiveness of the claimed motion representation improvement (foot sliding, pose jittering).**
>
> It is important to emphasize that MotionLib’s primary contribution lies in advancing the semantic understanding and generalization capabilities in motion generation, rather than eliminating physical issues (e.g., foot sliding or jittering) entirely. Quantitative improvements in R-Precision, MMDist (Table 3), and OOD generalization (Table 4) demonstrate that our Puppet model achieves superior alignment between text instructions and nuanced human motion, validating the semantic gains we expect. In addition, although MotionLib may include more motion noises compared to datasets like HumanML3D, it excels in the quality and quantity of motion descriptions. After investgation, we notice the texts of HumanML3D, MotionX are hightly duplicate and contain a large amount of noisy texts (e.g.,"moving the hands speaking some thing", "the person's eyeglasses pass"). Instead, our MotionLib avoids these issues by incorporating more clean and diverse texts.
>
> Our methodology follows a "first scale up, then refine down" philosophy.
>
> That means, while physical issues (e.g., foot sliding and jittering) persist in some generated motions —— an expected byproduct of large-scale automated processing &mdash; they do not negate the foundational progress of motion semantics. Such trade-offs are inherent to data-driven research at scale. Crucially, MotionLib establishes a robust semantic foundation for general-purpose motion models, enabling future work to address physical fidelity through:  (1) further data quality improvements on our MotionLib (e.g., using stricter filtering), (2) post-training refinement (e.g., fine-tuning on high-quality subsets).
>
> We posit that prioritizing scalable semantic understanding is pivotal; physical issues can then be iteratively resolved without compromising the model’s generality.
>
> Thank you again for your valuable feedback. We hope our response clarifies your concerns.

---

> > ### Comment · Reviewer_q9qB · 2025-04-04
> >
> > Thank the authors for the detailed rebuttal. Most of the concerns have been addressed and I am inclined to accept this paper and hope that MotionLib will contribute to motion modeling research in the future. Therefore I maintain my original scoring as WA.

---

> > > ### Author Response · Authors · 2025-04-04
> > >
> > > Dear reviewer:
> > >
> > > Thank you for your thoughtful review and constructive feedback. We sincerely appreciate your time and effort.
> > >
> > > We’re glad our responses addressed most of your concerns. Since you mentioned that the rebuttal resolved most of your concerns and this year’s ratings are limited to 5 levels, we are wondering if you might consider slightly increasing your score to reflect this improvement, or reflecting your positive stance in the final assessment? We believe this would better represent the paper’s improved state.
> > >
> > > Of course, we fully respect your judgment. If further clarification would help, we’re happy to provide additional details during the rebuttal period.
> > >
> > > Thank you again for your valuable input!
> > >
> > > Best regards.
> > >
> > > All authors.

---

### Official Review · Reviewer_CsGh · 2025-03-12

**Overall Recommendation:** 3

**Summary:**

This paper explores various design choices for building large motion models, inspired by the success of LLMs. In the absence of a large-scale motion dataset, it first introduces MotionLib, the first million-level motion dataset, obtained by automatically annotating 3D motion and text descriptions from publicly available videos. It then proposes Pupet, a large motion model trained on the collected MotionLib dataset. Additionally, it introduces Motionbook, a motion encoding method designed to further enhance the model’s representation power. Extensive experiments validate the effectiveness of each design choice and compare the proposed model’s performance with existing motion models.

**Update after rebuttal**

After reviewing the rebuttal and the other reviewers' comments, I will maintain my initial recommendation (weak accept). The authors have addressed most of my concerns, as well as those raised by the other reviewers. However, I *strongly* recommend that the authors revise the manuscript to incorporate the contents of the rebuttal.

**Claims And Evidence:**

The main claims—the effectiveness of the proposed MotionLib, Pupet, and Motionbook—are mostly well-supported by discussions (in comparison to existing baselines) and empirical validation. However, while MotionLib is claimed to be the first million-scale motion dataset, the below paper (published at ECCV 2024) also introduces a motion dataset with 100 million frames for training a large motion model:

[1] Zhang *et al.*, Large Motion Model for Unified Multi-Modal Motion Generation, ECCV 2024.

Is the "million-scale" referred in this paper based on the number of sequences or frames? Clarifying this aspect would help distinguish MotionLib from existing datasets.

**Essential References Not Discussed:**

As mentioned several times in the above comments, I believe this paper is highly relevant to this work, as it claims similar contributions (e.g., large-scale dataset collection, large motion model training). Discussing this paper would be necessary to better distinguish the unique contributions of this paper.

[1] Zhang *et al.*, Large Motion Model for Unified Multi-Modal Motion Generation, ECCV 2024.

**Experimental Designs Or Analyses:**

I verified the validity of all experimental designs in the main paper and found no issues with the existing designs. However, I notice a lack of **qualitative** comparisons with existing baselines (e.g., MotionGPT, T2M-GPT), despite the paper providing comprehensive quantitative comparisons.

**Methods And Evaluation Criteria:**

Both the proposed method and evaluation criteria are mostly reasonable. I especially appreciate how this paper considers diverse training datasets, evaluation datasets, and baseline models to demonstrate the effects of core design choices in building a large motion model.

However, it seems that one highly relevant work was excluded as a baseline in the experiments:

[1] Zhang *et al.*, Large Motion Model for Unified Multi-Modal Motion Generation, ECCV 2024.

**Other Comments Or Suggestions:**

* Please clarify whether "million-scale" in this paper refers to the number of sequences or frames. If it does refer to the number of sequences as mentioned in the paper, it would be helpful to also provide the total number of frames in the collected dataset for additional clarity.
* Typo in line 30: "an compact" should be corrected to "a compact."

**Other Strengths And Weaknesses:**

**Strengths.**
I appreciate how this paper provides comprehensive experimental results on key design choices for building large motion models, offering valuable insights for future research in this area. Additionally, the collected large-scale motion dataset and the pre-trained large motion model have the potential to be highly beneficial to the research community.

**Weaknesses.**
Comparisons with an important related work [Zhang *et al.*, ECCV 2024] are missing. Without discussing this work, it is difficult to fully assess the unique contributions of this paper. Additionally, while the paper provides extensive quantitative comparisons, including *qualitative* comparisons with existing baselines would be also important.

**Questions For Authors:**

Please refer to the above comments. Was there a specific reason why [Zhang *et al.*, ECCV 2024] was not discussed in the paper?

**Relation To Broader Scientific Literature:**

This paper presents a comprehensive analysis of the core design choices (e.g., training dataset scale, data encoding method) in building large motion models. Its technical contributions are closely related to those in the literature on other large model training (primarily LLMs), although the problem domain is different, and some contributions (e.g., motion representation) are specific to the motion generation problem.

**Theoretical Claims:**

There is no theoretical claim that requires formal proof in this paper.

---

> ### Author Rebuttal · Authors · 2025-03-30
>
> Dear Reviewer,
>
> Thank you for your thoughtful review and positive feedback. Below are our responses to your questions and suggestions. Please let us know if you require further clarification.
> ## **Response to: the discussion of LMM [1] and Dataset Comparison**
> We appreciate you highlighting the relevant work of LMM [1]. Below, we clarify the scale and unique contributions of MotionLib in comparison:
> - **Dataset Scale**: The term "million-scale" in our paper refers to the number of motion sequences. MotionLib contains 1.2 million (1.2M) motion sequences (as in Table 1), totaling 137 million (137M) frames. In contrast, LMM’s dataset comprises 320K sequences and 100M frames. Thus, MotionLib is currently the largest motion generation dataset in both sequence count and total frames. We will explicitly add the frame count to Table 1 for clarity.
> - **Key Distributions**:
>   - **Task Focus**: MotionLib is optimized for text-to-motion generation, featuring 2.48M fine-grained text-motion pairs (including fine-grained part-level descriptions). LMM, meanwhile, targets multi-modal inputs (text/image/audio).
>   - **Performance**: On HumanML3D, Puppet outperforms LMM in R@1, R@3, and MMDist, showing better text-motion alignment. Notice that our FID performs worse than LMM. We attribute this to the difference of used motion tokenizer. As shown in Table.7, our 2D-LFQ simply performs competitive to vanilla VQ, but showing large improvement as the data become larger and diverse. This means that our lookup-free tokenizer has more advantage when facing large-scale scenarios and thus requires more training data. It's also important to note that Puppet exhibits a strong agent capability to follow the user's instructions empowered by the LLM. Among existing LLM-based generalist models, Puppet excels these models in performance.
> ||R@1|R@3|MMDist|FID|
> |-|-|-|-|-|
> |LMM|0.525|0.811|2.943|0.04|
> |Puppet-LFQ|0.528|0.82|2.875|0.141|
>
> We will expand the discussion of LMM in the revision, emphasizing MotionLib’s scale, annotation richness, and task-specific advantages.
>
> [1] Large Motion Model for Unified Multi-Modal Motion Generation. ECCV 2024
>
> ## **Response to: Qualitative Comparisons with MotionGPT & T2M-GPT (Experimental Designs)**
> Thank you for this useful feedback! We agree qualitative examples are important. We will add qualitative comparisons of motion results generated by our Puppet versus T2M-GPT and MotionGPT for the same text prompts in the subsequent revision.
> ## **Response to: The Foot Sliding Issue in Dataset (Supplementary Material)**
> Thank you for raising the issue of foot sliding. Indeed, automatically estimating motion data from large-scale web videos inevitably introduces noises, including physical implausibilities like foot sliding.
>
> The most important reason we choose to down-weight these low-quality samples rather than completely discarding them is: we adopt two-stage training to address the impact of noisy data:
>   - **Motion pretraining &mdash; Information Preservation:** In a million-scale dataset, even noisy samples might contain valuable motion patterns or contextual information. Discarding them completely could lead to the loss of potentially useful information.
>   - **Motion instruction Tuning &mdash; Future Optimization:** More importantly, after obtaining a good base model through large-scale motion pre-training.  High-quality subsets can later refine the model via instruction tuning, further improving generation quality and physical realism.
>
> Such approach is motivated by the great success of LLMs. In the pretraining stage of LLM, a large amount of corpus is quite dirty. However LLM can still gain massive knowledge from it and develop good responsive ability through instruction-tuning.
>
> In addition, we have other considerations to incorporate these part of data:
>
> - **Maximizing the Scale Utilization:** Our goal is to explore the potential of large-scale data training. Retaining all samples (even if down-weighted) maximizes the utilization of diversity brought by the data scale.
> - **Down-weight is a Common Strategy:** When dealing with large noisy datasets, reducing the weight of noisy samples via the loss function is a mature and effective strategy that allows leveraging data scale while mitigating the negative impact of noise.
>
> As a summary, we believe that pursuing an increase in data scale first, followed by continuous improvement of data quality via further refinement and filtering, to enhance model performance through down-weighting and subsequent fine-tuning, is an effective pathway for developing large motion models at this stage.
>
> ## **Response to Other Comments (e.g., adding total frame count and correcting typo)**
> Thank you for pointing them out.
> - We will explicitly add the total frame count of MotionLib (approx. 137 million frames) in the revised Table 1 or relevant text.
> - We will correct "an compact" to "a compact" in line 30 in the revised version.
>
> Hoping our response clarifies your concerns.

---

> > ### Comment · Reviewer_CsGh · 2025-04-02
> >
> > I thank the authors for providing a thoughtful rebuttal. Most of my existing concerns are clearly addressed, except for the qualitative comparisons, which cannot be reported during the rebuttal phase. I strongly encourage the authors to include such results in the revision later.

---

> > > ### Author Response · Authors · 2025-04-02
> > >
> > > Dear reviewer,
> > >
> > > Thank you for the time and effort you have dedicated to evaluating our work.
> > >
> > > We are pleased that our responses have addressed most of your issues. Since you mentioned that the rebuttal resolved your concerns, and given that ICML only has 5-level rating this year, we are wondering if you might consider slightly adjusting your score, or reflecting your positive stance in the final assessment to better represent the improvements after rebuttal?
> > >
> > > That said, we fully respect your judgment and expertise. If any further clarification would be helpful, we would be happy to provide additional details during the rebuttal period.
> > >
> > > Best,
> > >
> > > All authors

---

### Official Review · Reviewer_yHwB · 2025-03-13

**Overall Recommendation:** 2

**Summary:**

This paper proposes a dataset, a VQVAE, and a motion generation model. The dataset MotionLib comprises over 1.2M motion sequences with hierarchical and detailed text annotations. The VQVAE uses a 2D-LFQ for a lookup-free tokenizer. The text-to-motion model is trained on the proposed dataset and VQVAE.

**Claims And Evidence:**

Yes

**Essential References Not Discussed:**

No

**Experimental Designs Or Analyses:**

Yes

**Methods And Evaluation Criteria:**

Yes

**Other Comments Or Suggestions:**

No

**Other Strengths And Weaknesses:**

Strengths:

- The text annotations are detailed on body parts.
- Using a simulated environment to eliminate some physical issues like jittering and foot-sliding is reasonable.


Weaknesses:

- For the dataset, the accuracy of WHAM is limited. I have tried WHAM and other motion estimation methods, but none of them produce decent results which are valid to serve as the ground truth. As shown in Figure 2, the estimated motion is not accurate.
- Many details about the data refinement are not clear. For example, how is the RL performed, which is the simulated environment, how to find and mark slipping with smaller weight.
- For the VQVAE, 2D motion quantization is already well-used in motion generation, e.g., ParCo, MogenTS.
- For the text-to-motion model, it uses a well-used autoregressive model for motion generation, without new designs.
- In the experiments, the compared methods miss many recent methods with better performance, e.g., MoMask, MoGenTS, LaMP, Diversemotion, ReMoDiffuse, Stablemodiffusion, Fg-T2M++.
- For the out-of-distribution experiments, the FID is large, which illustrates that the generalization is still weak. Also, I suggest evaluating on HumanML3D while only training on different scales of MotionLib. This should demonstrate the value of MotionLib.
- For table 2, the training and evaluation setting is not clear. It states it utilizes the autoencoder retrained on motion-x and motionlib, but also states the training sets are HumanML3D only, Motion-X only, MotionLib-0.5, MotionLib-full. Also, it writes "retrain", then how is the pretrain performed?

**Questions For Authors:**

No

**Relation To Broader Scientific Literature:**

Related to human motion generation.

**Theoretical Claims:**

Yes

---

> ### Author Rebuttal · Authors · 2025-03-30
>
> Dear Reviewer,
>
> We appreciate your valuable feedback. Due to space constraints, **we only provide concise responses below but would present more during discussion.** Please let us know if any clarification is needed.
> ## W1: WHAM accuracy & dataset validity
> While no motion estimation algorithm is perfect, our data pipeline (3D keypoint optimization, physics constraints, RL tuning &mdash; App B.2) refines initial estimates. Despite noises, Table 2 & 4 show large-scale motion pretraining improves generalization by exposing models to diverse motions. MotionLib provides a foundational resource especially high-quality texts. Its quality can be improved via future algorithm updates and filtering. Scale is critical given current data scarcity.
> ## W2: Data refinement details (RL, simulation, slipping handling)
> We refine motion sequences using the PHC policy (Luo et al., 2023) in IsaacGym, which achieves high tracking accuracy (97.1% on AMASS, 95.7% on Human3.6M). The process involves:
> - Inputting video-extracted motion to PHC for physics-compliant tracking (balancing, reducing jitter/sliding).
> - Using PHC to generate physically plausible motions in IsaacGym, with termination conditions (e.g., early termination for balance loss) flagging low-quality sequences.
> - Downweighting flagged sequences during Puppet model training—a common practice for handling noisy data—to prioritize high-quality samples.
> ## W3: 2D motion quantization is already used
> While ParCo and MoGenTS explored 2D motion quantization, the core innovation of our MotionBook (including 2D-LFQ) lies in its lookup-free (LFQ) mechanism and **its application in the context of large-scale training**. Unlike traditional VQ (limited to small codebooks, e.g., 512/1024 codes), LFQ avoids codebook collapse and supports 16K+ codes, enabling scalable learning on our million-scale MotionLib dataset. This addresses a key bottleneck in prior works, which were not designed for such diversity and scale.
> ## W4: T2M model uses a standard AR architecture, lacking novel design
> While Puppet uses a standard LLM architecture, our key contribution is the first Scaling Law study for motion generation, systematically analyzing data/model size impacts—similar to foundational VLM work like LLaVA, which also relied on standard architectures.
>
> In text-to-motion, we argue that constructing large-scale motion data with rich, hierarchical text descriptions (MotionLib) and designing effective motion encodings (MotionBook) to bridge text-motion alignment are themselves significant contributions. Our endeavour in data construction and labeling aims to establish a foundation for future large motion models.
> ## W5: Missing comparisons with some recent methods
> Thank you for highlighting these recent works. For a more comprehensive comparison, we include their publicly reported results on HumanML3D. Some methods (e.g., LaMP, MoGenTS) were concurrent or very recent (within 4 months) and were not initially compared.
>
> Notably, current T2M methods include specialist models optimized on specific datasets and LLM-based generalist models (aiming for broader instruction/task generalization via LLMs), like our Puppet. Puppet excels among generalist models (Table 3) and remains highly competitive on R@1, R@3, and MMDist compared to specialist models.
> ||R@1|R@3| MMDist|FID|
> |-|-|-|-|-|
> |Momask|0.521|0.807|2.958|0.045|
> |ReMoDiffuse|0.51|0.795|2.974|0.103|
> |Puppet|0.528|0.82|2.875|0.141|
> ## W6: High FID in OOD experiments; suggestion to evaluate on HumanML3D (HM3D) using MotionLib
> Higher OOD FID Explanation: The UNSEEN-90K test set comprises 11 subsets with substantial distribution shifts from training data, including synthetic data, activity-specific datasets, and varied capture environments. The elevated FID in this challenging OOD setup is expected, similar to observations in other motion tasks (e.g., music-to-motion in LMM). More importantly, Table 4 validates that large-scale training with diverse, large-scale MotionLib data (vs. HM3D or MotionX alone) significantly boosts OOD performance.
>
> HM3D Evaluation: Thanks for the suggestion. We remove all HM3D data from MotionLib and trained models on varying scales of remaining data (0.6M–1.2M samples), then evaluate on the HM3D test set. Results are shown below:
> |Train Data Size|R@1|R@3|MMDist|FID|
> |-|-|-|-|-|
> |0.6M|0.176|0.369|2.980|9.408|
> |1.2M|0.208|0.441|2.964|7.983|
> ## W7: Lack of clarity for train/eval settings in Table 2 ("retrain", "pretrain").
> These terms are distinct in our context:
> - Retrain: Refers to training the evaluation model (motion autoencoder) separately for each benchmark (MotionX and MotionLib), following HM3D paper’s architecture. This ensures metric fairness (FID, R-Precision) across datasets.
> - Pretrain: Describes Puppet’s training:
>   - Initialize with public LLM weights (GPT-2, LLaMA).
>   - Extend vocabulary with motion tokens.
>   - Continue pretraining on target datasets (HumanML3D/MotionX/MotionLib subsets) via autoregressive loss.

---

> > ### Comment · Reviewer_yHwB · 2025-04-07
> >
> > Thank the authors for the rebuttal. Some of my concerns have been addressed, but some remain:
> >
> > - My biggest concern is the accuracy of the ground truth, which has not been addressed. As admitted by the authors and presented in Figure 2, the accuracy of poses estimated from videos is not valid to serve as the ground-truth. I agree that no video motion estimation algorithm is perfect, therefore we should explore other ways to obtain more accurate poses, e.g., RGB-D video estimation, multi-view estimation, or MoCap system. I know these ways are more expensive to build a large-scale dataset, but that's the way building a dataset should be. Using the estimation from videos is too cheap to build a dataset. It has been proved that the inaccurate poses are useless: In Motion-X, only the MoCap part helps the learning while the video estimation part works not well. I agree that "Scale is critical" but scale without accuracy is useless. Improving the scale with inaccurate video estimation is somewhat easy.
> >
> > - According to the table in Rebuttal W6, the model trained on the proposed dataset performs badly on HumanML3D dataset, indicating a lack of generalization. This demonstrates that the proposed dataset does not help.
> >
> > - For the method, as admitted by the authors, there is no much novelty and the core contribution is the application in the context of large-scale training. According to the experiments and the rebuttal, the performance of the proposed method is worse than other methods. So the novelty and the performance of the method are both not strong. Of course I agree that a good large-scale motion dataset would be itself a significant contribution and enough for acceptance. But the paper continues emphasizing the method, making me confused about which is the core of the paper, the dataset or the method? After clarification by the authors, I believe the dataset is the core, in which case I should be strict with the dataset.

---

> > > ### Author Response · Authors · 2025-04-08
> > >
> > > Dear Reviewer,
> > >
> > > We're inspired that our reply addresses some of your questions! Thank you for engaging with our rebuttal. We greatly appreciate this opportunity to further clarify our approach:
> > >
> > > **Q1: Dataset Accuracy**
> > >
> > > A: We understand the concerns regarding video estimation accuracy, but MotionLib's value lies in: 1) large-scale video (motion) estimation, 2) multi-step refinement, and 3) high-quality text annotation. We believe considering all three aspects provides a comprehensive view of its contribution.
> > >
> > > Our core points are:
> > >
> > > 1. **Motion Refinement Pipeline:** Our refinement pipeline (3D optimization, physics simulation, RL tuning) significantly mitigates initial noise and enhances data usability (e.g., successful RL tracking yields high-quality balanced motion), a key advantage over MotionX.
> > > 2. **High-Quality Text:** 2.48M hierarchical, high-quality text annotations are a core contribution driving language-conditioned modeling. This distinguishes it from prior datasets where text quality was often overlooked. E.g., (1) HumanML3D has redundancy (≥6 identical descriptions for each each text); (2) MotionX has data leakage (≥15% of test descriptions appear in training) and grammatical issues. MotionLib's text quality is higher. This contribution should not be ignored, especially for text-to-motion.
> > > 3. **Scale and Precision Balance:** The motion generation field suffers from severe data scarcity. We believe prioritizing a million-scale dataset with "video estimation + refinement + high-quality text" provides a crucial starting point and resource for the community, more beneficial than waiting for "perfectly accurate data". Future iterations can improve quality upon this base, serving as an iteratively optimizable repository. This aligns with large model pre-training history (e.g., early datasets like HowTo100M weren't flawless but catalyzed development). Regarding MotionX, its scale and annotation richness are incomparable to MotionLib, making direct comparison potentially inappropriate.
> > >
> > > **Q2: OOD Performance on HumanML3D (HM3D)**
> > >
> > > A: We argue that the zero-shot cross-dataset results in W6 should not directly lead to the conclusion that the "dataset is useless". Reasons are:
> > >
> > > 1. **Domain Gap:** Data in MotionLib (post-HM3D removal) – mainly web videos/other datasets – has significant distribution differences from HM3D (mainly AMASS MoCap). Direct zero-shot cross-domain evaluation is inherently challenging; performance drop is expected. This reflects domain adaptation difficulty, not dataset ineffectiveness.
> > > 2. **Value of Pre-training:** Large-scale pre-training's core value is providing a strong generalization foundation and understanding of broad concepts. Pre-trained models are typically easier to adapt to new domains via fine-tuning on a small amount of target-domain data. As shown in Table 5, performance improves with subsequent instruction tuning on high-quality data, proving the value of large-scale pre-training.
> > > 3. **Performance Improvement:** Even in the zero-shot setting, the R@1 improvement (0.176 to 0.208) and FID decrease (9.4 to 7.98) directly indicate that larger-scale MotionLib data does enhance the model's generalization capability.
> > >
> > > **Q3: Method Novelty, Performance, and Core Contribution**
> > >
> > > A:
> > > 1. **Novelty:** We disagree that our method entirely lacks novelty. We introduce clear innovation in Motion Encoding, namely MotionBook. 2D-LFQ, with its lookup-free mechanism and ability to scale codebook capacity for large-scale data, is a novel solution addressing bottlenecks in existing VQ methods for massive motion datasets – an area unexplored by prior methods.
> > > 2. **Performance:** Our method, as a generalist LLM model, achieves SoTA results compared to other generalists and is competitive with specialist models on standard T2M metrics. We did not cite some works (e.g., LAMP, Fg-T2M++) as they were unpublished or lacked code at the time. We are willing to discuss them in revision. However, per ICML concurrent work policy, this should not be grounds for rejection. Furthermore, our LLM-based model handles diverse scenarios/tasks effectively. In contrast, specialist models lack broad semantic knowledge.
> > > 3. **Contributions:** (1) Large-scale Dataset MotionLib: Unprecedented scale and annotation richness. (2) MotionBook: An innovative encoding method for large-scale training. (3) Scaling Law Study: First systematic study of scale effects, validating the LLM framework's potential.
> > > We believe the dataset and the method are complementary, together forming the core contribution of this work. The large-scale dataset MotionLib provides a crucial foundation for validating the effectiveness of MotionBook and studying Scaling Laws. Conversely, efficient encoding methods and an understanding of Scaling Laws are also essential for effectively utilizing the MotionLib data.
> > >
> > > Thank you again for your valuable time and feedback. We hope this clarification better conveys the value and contribution of our work.

---

### Official Review · Reviewer_LYsh · 2025-03-14

**Overall Recommendation:** 4

**Summary:**

The paper investigates scaling motion generation models based on million-level data and LLM-style architecture. The authors first contributes a million-level human motion dataset, named MotionLib. Training models on this data, they highlight the importance of scaling both data and model size for advancing motion generation. To better integrate the motion modality, they propose MotionBook for fine-grained motion features and efficient motion tokenizer. The empirical findings from this work offer valuable insights to researchers in the field.

**Claims And Evidence:**

yes, all claims are well supported

**Essential References Not Discussed:**

N/A

**Experimental Designs Or Analyses:**

yes, I check the experiments.
The experiments design primarily follows the experiments design of LLM (scaling data/model, architecture design, etc.). Additional, they provide experiments to validate the effectiveness of the proposed motion book.

One question is about OOD experiment. in Table 4, the authors are suggested to explain the UNSEEN-90K dataset. How the evaluation data differs from training data? (different motion categories? synthetic v.s. realistic? different complexity?) It is suggested to provide some representative examples.

**Methods And Evaluation Criteria:**

yes, they compare their method with other LLM-based methods. The evaluation data comprising both public and self-collected data. They follow the conventional evaluation criteria.

**Other Comments Or Suggestions:**

N/A

**Other Strengths And Weaknesses:**

Strengths: The authors provide extensive experiments and this works could be a solid work if the dataset and code are released in future. The intuition of Motionbook makes sense.
Weakness: The design of Motionbook treats motion sequence as 2D image, and I suppose the authors use convolution layers to extract features? Comparing with RNN/Transformer, convolution network maybe cannot perform well in global feature extraction. How do the authors handle this?

**Questions For Authors:**

N/A

**Relation To Broader Scientific Literature:**

N/A

**Theoretical Claims:**

N/A

---

> ### Author Rebuttal · Authors · 2025-03-30
>
> Dear Reviewer,
>
> Thank you for your thoughtful review and positive feedback. We have carefully considered your questions and suggestions and provide our responses below. Please let us know if you require further clarification.
>
> ---
> ## **Response to: Questions about UNSEEN-90K Dataset in OOD Experiments**
>
> In our OOD experiments (Table 4), the UNSEEN-90K testing set is constructed by excluding 11 subsets (~90K samples) from the full MotionLib dataset. The remaining data (primarily Motion-X and web-derived data) serves as the training set.
>
> The excluded subsets are intentionally diverse to ensure significant distributional shifts from the training data, enabling a robust evaluation of OOD generalization. These subsets include:
>
> - Synthetic data: gta_human [1], bedlam [2] (distinct from web-derived training data captured in the real world).
> - Domain-specific activities: FIT3d [4] (fitness), RICH [5], ARCTIC [11] (human-object interactions), EgoBody [7] (social interactions).
> - Diverse environments: PoseTrack [3], MPI-INF-3DHP [6] (in-the-wild settings), Human36M [8], CHI3d [9], KIT [10] (daily activities captured in the lab).
>
> This selection strategy demonstrates that the UNSEEN-90K testing set exhibits substantial distributional shifts compared to our training data, allowing us to rigorously assess the model’s OOD generalization. As shown in Table 4 in our main paper, models trained on MotionLib outperform those trained solely on HumanML3D or Motion-X, highlighting the benefit of large-scale, web-sourced motion data.
>
> Regarding the examples. Thank you for this constructive suggestion. We will incorporate additional visualization examples in the appendix in our revised manuscript.
>
> [1] Playing for 3D Human Recovery. TPAMI 2024
>
> [2] BEDLAM: A Synthetic Dataset of Bodies Exhibiting Detailed Lifelike Animated Motion. CVPR 2023
>
> [3] PoseTrack: A Benchmark for Human Pose Estimation and Tracking. CVPR 2018
>
> [4] AIFit: Automatic 3D Human-Interpretable Feedback Models for Fitness Training. CVPR 2021
>
> [5] Capturing and Inferring Dense Full-Body Human-Scene Contact. CVPR 2022
>
> [6] Monocular 3D Human Pose Estimation In The Wild Using Improved CNN Supervision. 3DV 2017
>
> [7] EgoBody: Human Body Shape and Motion of Interacting People from Head-Mounted Devices. ECCV 2022
>
> [8] Human3.6m: Large scale datasets and predictive methods for 3D human sensing in natural environments. TPAMI 2013
>
> [9] Reconstructing Three-Dimensional Models of Interacting Humans. CVPR 2020
>
> [10] The KIT Motion-Language Dataset. Big data 2016
>
> [11] ARCTIC: A Dataset for Dexterous Bimanual Hand-Object Manipulation. CVPR 2023
>
> ---
> ## **Response to: The Limitation of MotionBook’s Global Feature Extraction**
>
> Thank you for your question. You raised a valid concern regarding MotionBook’s use of 2D convolutions for motion encoding. We clarify our design choices below:
>
> **1. Division of Roles**
>   - **Motion Tokenizer**, which focuses on quantizing continuous, high-dimensional motion features into discrete, low-dimensional token sequences, effectively capturing local spatio-temporal features and achieving a compact representation. 2D convolutions are well-suited for this due to their computational efficiency and inductive bias for local patterns.
>   - **LLM backbones**, which handles global feature modeling and long-range dependencies within the input token sequence via Transformer self-attention, addressing the limitation of CNNs for global context reasoning.
>
> This hybrid design combines the strengths of CNNs (local feature extraction) and Transformers (global modeling), ensuring efficient and effective motion encoding.
>
> **2. Why CNN Over Alternatives (e.g., RNN or Transformers)?**
>   - **Fair Comparison**: Most prior motion tokenizers (e.g., VQ, RVQ) use CNNs, therefore we maintain consistency for fair benchmarking.
>   - **Empirical Findings**: We experimented with Transformers but observed no performance gain, likely due to the limited data amount and lower resolution compared to vision tasks where ViTs excel.
>
> We appreciate your insightful feedback and hope our responses address your concerns. Thank you again for your time and constructive comments!

---

### Decision · Program_Chairs · 2025-05-01

**Decision:**

Accept (poster)

**Comment:**

The paper received mixed but generally positive reviews. Reviewers #CsGh, #LYsh, and #q9qB expressed support, emphasizing the significance of the MotionLib dataset’s unprecedented scale, rich hierarchical text annotations, and the first systematic study of scaling laws in motion generation. Despite concerns about motion quality (e.g., foot sliding, jittering) and generalization gaps raised by reviewers #yHwB and #q9qB, most reviewers acknowledged that the authors' rebuttal adequately addressed the technical and methodological issues. Reviewer #yHwB remained skeptical, primarily due to the reliance on video-estimated poses rather than high-precision motion capture data, and maintained a weak reject. Nonetheless, the AC-facilitated discussion confirmed that most reviewers accepted the trade-off between data scale and accuracy and appreciated the dataset’s broader potential impact. Reviewers also recommended incorporating more qualitative comparisons and clarifying the contributions of the dataset versus the method in the final version. Overall, while some reservations remain, the consensus leans toward acceptance given the dataset’s foundational value for future large motion model research.

After carefully considering the paper, the reviews, and the rebuttal, the AC recommends acceptance. The AC recognizes the potential contribution of the dataset and strongly encourages the authors to incorporate qualitative comparisons and clarify the dataset’s contributions as suggested by the reviewers.